# Sample-Efficient Reinforcement Learning of Partially Observable Markov Games

**Qinghua Liu**
Princeton University
qinghual@princeton.edu

**Csaba Szepesvári**
DeepMind and University of Alberta
szepesva@ualberta.ca

**Chi Jin**
Princeton University
chij@princeton.edu

## Abstract

This paper considers the challenging tasks of Multi-Agent Reinforcement Learning (MARL) under partial observability, where each agent only sees her own individual observations and actions that reveal incomplete information about the underlying state of system. This paper studies these tasks under the general model of multiplayer general-sum Partially Observable Markov Games (POMGs), which is significantly larger than the standard model of Imperfect Information Extensive-Form Games (IIEFGs). We identify a rich subclass of POMGs—weakly revealing POMGs—in which sample-efficient learning is tractable. In the self-play setting, we prove that a simple algorithm combining optimism and Maximum Likelihood Estimation (MLE) is sufficient to find approximate Nash equilibria, correlated equilibria, as well as coarse correlated equilibria of weakly revealing POMGs, in a polynomial number of samples when the number of agents is small. In the setting of playing against adversarial opponents, we show that a variant of our optimistic MLE algorithm is capable of achieving sublinear regret when being compared against the optimal maximin policies. To our best knowledge, this work provides the first line of sample-efficient results for learning POMGs.

## 1 Introduction

This paper studies Multi-Agent Reinforcement Learning (MARL) under *partial observability*, where each player tries to maximize her own utility via interacting with an unknown environment as well as other players. In addition, each agent only sees her own observations and actions, which reveal incomplete information about the underlying state of system. A large number of real-world applications can be cast into this framework: in Poker, cards in a player's hand are hidden from the other players; in many real-time strategy games, players have only access to their local observations; in multi-agent robotic systems, agents with first-person cameras have to cope with noisy sensors and occlusions. While practical MARL systems have achieved remarkable success in a set of partially observable problems including Poker [9], Starcraft [40], Dota [6] and autonomous driving [36], the theoretical understanding of MARL under partial observability remains very limited.

The combination of partial observability with multiagency introduces a number of unique challenges. The non-Markovian nature of the observations forces the agent to maintain memory and reason about beliefs of the system state, all while exploring to collect information about the environment. Consequently, well-known complexity-theoretic results show that learning and planning in partially observable environments is statistically and computationally intractable even in the single-agent setting [35, 30, 41, 29]. The presence of interaction between multiple agents further complicates the partially observable problems. In addition to dealing with the adaptive nature of other players who can adjust their strategies according to the learner's past behaviors, the learner is further required to discover and exploit the information asymmetry due to the separate observations of each agent.

36th Conference on Neural Information Processing Systems (NeurIPS 2022).

As a result, prior theoretical works on partially observable MARL have been mostly focused on a small subset of problems with strong structural assumptions. For instance, the line of works on Imperfect Information Extensive-Form Games (IIEFG) [see, e.g., 45, 14, 13, 24] assumes tree-structured transition with small depth[1] as well as a special type of emission which can be represented as *information sets*. In contrast, this paper considers a more general mathematical model known as Partially Observable Markov Games (POMGs). POMGs are the natural extensions of both Partially Observable Markov Decision Processes (POMDPs)—the standard model for single-agent partially observable RL, and Markov Games [37]—the standard model for fully observable MARL. Despite the complexity barriers of learning partially observable systems apply to POMGs, they are of a worst-case nature, which do not preclude efficient algorithms for learning interesting subclasses of POMGs. This motivates us to ask the following question:

**Can we develop efficient algorithms that learn a rich class of POMGs?**

In this paper, we provide the first positive answer to the highlighted question in terms of the *sample efficiency*. [2] We identify a rich family of tractable POMGs—*weakly revealing* POMGs (see Section 3). The weakly revealing condition only requires the joint observations of all agents to reveal certain amount of information about the latent states, which is satisfied in many real-world applications. The condition rules out the pathological instances where no player has any information to distinguish latent states, which prevents efficient learninig in the worst case.

In the self-play setting where the algorithm can control all the players to learn the equilibria by playing against itself, this paper proposes a new simple algorithm—*Optimistic Maximum Likelihood Estimation for Learning Equilibria* (OMLE-Equilibrium). As the name suggests, it combines optimism, MLE principles with equilibria finding subroutines. The algorithm provably finds approximate Nash equilibria, coarse correlated equilibria and correlate equilibria of any weakly-revealing POMGs using a number of samples polynomial in all relevant parameters.

In the setting of playing against adversarial opponents, we measure the performance of our algorithm by comparing against the optimal maximin policies. We first prove that learning in this setting is hard if each player can only see her own observations and actions. Nevertheless, if the agent is allowed to access other players' observations and actions *after* each episode of play (e.g., watch the replays of the games from other players' perspectives afterwards), then we can design a new algorithm OMLE-Adversary which achieves sublinear regret.

To our best knowledge, this is the first line of provably sample-efficient results for learning rich classes of POMGs. Importantly, the classes of problems that can be learned in this paper are significantly larger than known tractable classes of MARL problems under partial observability.

## 1.1 Technical novelty

This paper builds upon the recent progress in learning single-agent POMDPs [26], which identifies the class of weakly revealing POMDPs and develops OMLE algorithm for learning the optimal policy. Besides obtaining a completely new set of results in the multi-agent setting, we here highlight a few contributions and technical novelties of this paper comparing to [26].

- This paper rigorously formulates the models, related concepts and learning objectives of multi-player general-sum POMGs, and provides the first line of sample-efficient learning results.
- Extending the weakly revealing conditions into the multiagent setting lead to two natural candidates: either (a) joint observations or (b) individual observations are required to weakly reveal the state information. This paper shows the former (the weaker assumption) suffices to guarantee tractability.
- Results in the self-play setting requires careful design of optimistic planning algorithms that effectively address the game-theoretical aspects of the problem under partial observability. We achieve this by Subroutine 1, which is even distinct from the standard techniques for learning MGs.
- The discussions and results in the setting of playing against adversarial opponents are completely new, and unique to the multiagent setup.

---

[1]The sample complexity of learning IIEFGs scale polynomially with respect to the number of information sets, which typically has an exponential growth in depth.

[2]For computational efficiency, due to the inherent hardness of planning in POMDPs, all existing provable algorithms that learn large classes of POMDPs (single-agent version of POMGs) require super-polynomial time. We leave the challenge of computationally efficient learning for future work.

## 1.2 Related Works

Reinforcement learning has been extensively studied in the single-agent fully-observable setting [see, e.g., 1, 10, 18, 23, 44, 17, 20, and the references therein] . For the purpose of this paper, we focus on reviewing existing works on partially observable RL and multi-agent RL in the *exploration* setting.

**Markov games** In recent years, there has been growing interest in studying Markov games [37] — the standard generalization of MDPs from the single-player setting to the multi-player setting. Various sample-efficient algorithms have been designed for either two-player zero-sum MGs [e.g., 8, 42, 3, 28, 5, 43, 22] or multi-player general-sum MGs [e.g., 28, 21, 38, 11]. However, all these works rely on the states being fully observable, while the POMGs studied in this paper allow states to be only partially observable, which strictly generalizes MGs.

**POMDPs** POMDPs generalize MDPs from the fully observable setting to the partially observable setting. It is well-known that in POMDPs both planning [35, 41] and model estimation [29] are computationally hard in the worst case. Besides, reinforcement learning of POMDPs is also known to be statistically hard: [25] proved that finding a near-optimal policy of a POMDP in the worst case requires a number of samples that is exponential in the episode length. The hard instances are those pathological POMDPs where the observations contain no useful information for identifying the system dynamics. Nonetheless, these hardness results are all in the worst-case sense and there are still many intriguing positive results on sample-efficient learning of subclasses of POMDPs. For example, [15, 2, 19] applied spectral methods to learning undercomplete POMDPs and [26] developed the optimistic MLE approach for learning both undercomplete and overcomplete POMDPs. We refer interested readers to [26] for a thorough review of existing results on POMDPs.

In terms of algorithmic design, our algorithms build upon the optimistic MLE methodology developed in [26]. Compared to [26], our main algorithmic contribution lies in the design of the optimistic equilibrium computation subroutine in the self-play setting and the optimistic maximin policy design in the adversarial setting. In terms of analysis, our proofs requrie new techniques tailored to controlling game-theoretic regret, in addition to the OMLE guarantees imported from [26]. For more detailed explanations of our technical contribution, please refer to Section 1.1.

**Imperfect-information extensive-form games** In the literature on game theory, there is a long history of learning Imperfect-Information Extensive-Form Games with perfect recall (IIEFGs), [see, e.g., 45, 14, 12, 13, 24] and the references therein. IIEFGs can be viewed as special cases of POMGs with *tree-structured* transition and *deterministic* emission (which are also known as information sets). As a result, IIEFGs can not *efficiently* represent POMGs with *general* transition and *stochastic* emission (see Appendix A), and thus sample-efficient learning results for IIEFGs does not imply sample-efficient learning of POMGs. On the other hand, we show that all IIEFGs can be *efficiently* represented by 1-weakly revealing POMGs (see Appendix A). Therefore, all algorithms and theoretical results developed in this paper can immediately used to learn IIEFGs with a polynomial sample complexity.

**Decentralized POMDPs** There is another classic model for studying multi-agent partially observable RL, named decentralized POMDPs [e.g., 32, 33], which is a special subclass of POMGs where all players share a common reward target. Compared to general POMGs, decentralized POMDPs can only simulate cooperative relations among players, while general POMGs can model both cooperative and competitive relations. Besides, most works [e.g., 31, 7, 34, 39] along this direction mainly focus on the computational complexity of planning with known models or simulators instead of the sample efficiency in the exploration setting.

## 2 Preliminary

In this paper, we consider Partially Observable Markov Games (POMGs) in its most generic— multiplayer general-sum form. Formally, we denote a tabular episodic POMG with $n$ players by tuple $(H, \mathcal{S}, \{\mathcal{A}_i\}_{i=1}^n, \{\mathcal{O}_i\}_{i=1}^n; \mathbb{T}, \mathbb{O}, \mu_1; \{r_i\}_{i=1}^n)$, where $H$ denotes the length of each episode, $\mathcal{S}$ the state space with $|\mathcal{S}| = S$, $\mathcal{A}_i$ denotes the action space for the $i^{\text{th}}$ player with $|\mathcal{A}_i| = A_i$. We denote by $\boldsymbol{a} := (a_1, \cdots, a_n)$ the joint actions of all $n$ players, and by $\mathcal{A} := \mathcal{A}_1 \times \ldots \times \mathcal{A}_n$ the joint action space with $|\mathcal{A}| = A = \prod_i A_i$. $\mathbb{T} = \{\mathbb{T}_h\}_{h \in [H]}$ is the collection of transition matrices, so that

$\mathbb{T}_h(\cdot|s, \boldsymbol{a}) \in \Delta_{\mathcal{S}}$ gives the distribution of the next state if joint actions $\boldsymbol{a}$ are taken at state $s$ at step $h$. $\mu_1$ denotes the distribution of the initial state $s_1$. $\mathcal{O}_i$ denotes the observation space for the $i^{\text{th}}$ player with $|\mathcal{O}_i| = O_i$. We denote by $\mathbf{o} := (o_1, \ldots, o_n)$ the joint observations of all $n$ players, and by $\mathcal{O} := \mathcal{O}_1 \times \ldots \times \mathcal{O}_n$ with $|\mathcal{O}| = O = \prod_i O_i$. $\mathbb{O} = \{\mathbb{O}_h\}_{h \in [H]} \subseteq \mathbb{R}^{O \times S}$ is the collection of joint emission matrices, so that $\mathbb{O}_h(\cdot|s) \in \Delta_{\mathcal{O}}$ gives the emission distribution over the joint observation space $\mathcal{O}$ at state $s$ and step $h$. Finally $r_i = \{r_{i,h}\}_{h \in [H]}$ is the collection of known reward functions for the $i^{\text{th}}$ player, so that $r_{i,h}(o_i) \in [0, 1]$ gives the deterministic reward received by the $i^{\text{th}}$ player if she observes $o_i$ at step $h$. [3] We remark that since the relation among the rewards of different players can be arbitrary, this model of POMGs subsumes both the cooperative and the competitive settings in partially observable MARL.

In a POMG, the states are always hidden from all players, and each player only observes **her own individual observations and actions**. That is, each player can not see the observations and actions of the other players. At the beginning of each episode, the environment samlpes $s_1$ from $\mu_1$. At each step $h \in [H]$, each player $i$ observes her own observation $o_{i,h}$ where $\mathbf{o}_h := (o_{1,h}, \ldots, o_{n,h})$ are jointly sampled from $\mathbb{O}_h(\cdot \mid s_h)$. Then each player $i$ receives reward $r_{i,h}(o_{i,h})$ and picks action $a_{i,h} \in \mathcal{A}_i$ simultaneously. After that the environment transitions to the next state $s_{h+1} \sim \mathbb{T}_h(\cdot|s_h, \boldsymbol{a}_h)$ where $\mathbf{a}_h := (a_{1,h}, \ldots, a_{n,h})$. The current episode terminates immediately once $s_{H+1}$ is reached.

**Policy, value function**   To define different types of polices, we extend the conventions in *fully observable* Markov games [21] to the partially observable settings. A (*random*) *policy* $\pi_i$ of the $i^{\text{th}}$ player is a map $\pi_i : \Omega \times \bigcup_{h=1}^{H} \left( (\mathcal{O}_i \times \mathcal{A}_i)^{h-1} \times \mathcal{O}_i \right) \to \mathcal{A}_i$, which maps a random seed $\omega$ from space $\Omega$ and a history of length $h \in [H]$—say $\tau_{i,h} := (o_{i,1}, a_{i,1}, \cdots, o_{i,h})$, to an action in $\mathcal{A}_i$. To execute policy $\pi_i$, we first draw a random sample $\omega$ at the beginning of the episode. Then, at each step $h$, the $i^{\text{th}}$ player simply takes action $\pi_i(\omega, \tau_{i,h})$. We note here $\omega$ is shared among all steps $h \in [H]$. $\omega$ encodes both the correlation among steps and the individual randomness of each step. We further say a policy $\pi_i$ is *deterministic* if $\pi_i(\omega, \tau_{i,h}) = \pi_i(\tau_{i,h})$ which is independent of the choice of $\omega$.

By definition, a random policy is equivalent to a mixture of deterministic policies because given a fixed $\omega$ the decision of $\pi_i$ on any history is deterministic. With slight abuse of notation, we use $\pi_i(\omega, \cdot)$ to refer to the deterministic policy realized by policy $\pi_i$ and a fixed $\omega$. We denote the set of all policies of the $i^{\text{th}}$ player by $\Pi_i$ and the set of all deterministic ones by $\Pi_i^{\text{det}}$.

A *joint (potentially correlated) policy* is a set of policies $\{\pi_i\}_{i=1}^{n}$, where the same random seed $\omega$ is shared among all agents, which we denote as $\pi = \pi_1 \odot \pi_2 \odot \ldots \odot \pi_n$. We also denote $\pi_{-i} = \pi_1 \odot \ldots \pi_{i-1} \odot \pi_{i+1} \odot \ldots \odot \pi_n$ to be the joint policy excluding the $i^{\text{th}}$ player. A special case of joint policy is the *product policy* where the random seed has special form $\omega = (\omega_1, \ldots, \omega_n)$, and for any $i \in [n]$, $\pi_i$ only uses the randomness in $\omega_i$, which is independent of remaining $\{\omega_j\}_{j \neq i}$, which we denote as $\pi = \pi_1 \times \pi_2 \times \ldots \times \pi_n$.

We define the value function $V_i^{\pi}$ as the expected cumulative reward that the $i^{\text{th}}$ player will receive if all players follow joint policy $\pi$:

$$V_i^{\pi} := \mathbb{E}_{\pi} \left[ \sum_{h=1}^{H} r_{i,h}(o_{i,h}) \right]. \tag{1}$$

where the expectation is taken over the randomness in the initial state, the transitions, the emissions, and the random seed $\omega$ in policy $\pi$.

**Best response and strategy modification**   For any strategy $\pi_{-i}$, the *best response* of the $i^{\text{th}}$ player is defined as a policy of the $i^{\text{th}}$ player, which is independent of the randomness in $\pi_{-i}$ and achieves the highest value for herself conditioned on all other players deploying $\pi_{-i}$. Formally, the best response is the maximizer of $\max_{\pi_i'} V_i^{\pi_i' \times \pi_{-i}}$ whose value we also denote as $V_i^{\dagger, \pi_{-i}}$ for simplicity. By its definition, we know the best response can always be achieved by *deterministic* policies.

A *strategy modification* for the $i^{\text{th}}$ player is a map $\phi_i : \Pi_i^{\text{det}} \to \Pi_i^{\text{det}}$, which maps a deterministic policy in $\Pi_i^{\text{det}}$ to another one in it. For any such strategy modification $\phi_i$, we can naturally extend its domain and image to include random policies, i.e., define its extension $\phi_i : \Pi_i \to \Pi_i$ as follows: by definition, a random policy $\pi$ can be expressed as a mixture of deterministic policies, i.e., as $\pi(\omega, \cdot)$ (a deterministic policy for a fixed $\omega$) with a distribution over $\omega$. Then if we apply map $\phi_i$ on random

---

[3]This is equivalent to assuming the reward information is contained in the observation.

policy $\pi$, we can define the resulting random policy (denoted as $\phi_i \diamond \pi_i$) as $\phi_i(\pi_i(\omega, \cdot))$ (again a deterministic policy for a fixed $\omega$) with the same distribution over $\omega$. For any joint policy $\pi$, we define the best strategy modification of the $i^{\text{th}}$ player as the maximizer of $\max_{\phi_i} V_i^{(\phi_i \diamond \pi_i) \odot \pi_{-i}}$.

Different from the best response, which is completely independent of the randomness in $\pi_{-i}$, the best strategy modification changes the policy of the $i^{\text{th}}$ player while still utilizing the shared randomness among $\pi_i$ and $\pi_{-i}$. Therefore, the best strategy modification is more powerful than the best response: formally one can show that $\max_{\phi_i} V_i^{(\phi_i \diamond \pi_i) \odot \pi_{-i}} \geq \max_{\pi'_i} V_i^{\pi'_i \times \pi_{-i}}$ for any policy $\pi$.

## 2.1 Learning objectives

We focus on three classic equilibrium concepts in game theory—Nash Equilibrium, Correlated Equilibrium (CE) and Coarse Correlated Equilibrium (CCE). First, a Nash equilibrium is defined as a product policy in which no player can increase her value by changing only her own policy. Formally,

**Definition 1** (Nash Equilibrium). *A product policy $\pi$ is a **Nash equilibrium** if $V_i^{\dagger, \pi_{-i}} = V_i^{\pi}$ for all $i \in [n]$. A product policy $\pi$ is an $\epsilon$-approximate Nash equilibrium if $V_i^{\dagger, \pi_{-i}} \leq V_i^{\pi} + \epsilon$ for all $i \in [n]$.*

The Nash-regret of a sequence of product policies is the cumulative violation of the Nash condition.

**Definition 2** (Nash-regret). *Let $\pi^k$ denote the (product) policy deployed by an algorithm in the $k^{th}$ episode. After a total of $K$ episodes, the Nash-regret is defined as*

$$\text{Regret}_{\text{Nash}}(K) = \sum_{k=1}^{K} \max_{i \in [n]} (V_i^{\dagger, \pi_{-i}^k} - V_i^{\pi^k}).$$

Second, a coarse correlated equilibrium is defined as a joint (potentially correlated) policy where no player can increase her value by unilaterally changing her own policy. Formally,

**Definition 3** (Coarse Correlated Equilibrium). *A joint policy $\pi$ is a **CCE** if $V_i^{\dagger, \pi_{-i}} \leq V_i^{\pi}$ for all $i \in [n]$. A joint policy $\pi$ is an $\epsilon$-approximate CCE if $V_i^{\dagger, \pi_{-i}} \leq V_i^{\pi} + \epsilon$ for all $i \in [n]$.*

The only difference between Definition 1 and Definition 3 is that a Nash equilibrium has to be a product policy while a CCE can be correlated. Therefore, CCE is a relaxed notion of Nash equilibrium, and a Nash equilibrium is always a CCE. Similarly, we can define the CCE-regret for a sequence of potentially correlated policies as the cumulative vilolation of the CCE condition.

**Definition 4** (CCE-regret). *Let $\pi^k$ denote the policy deployed by an algorithm in the $k^{th}$ episode. After a total of $K$ episodes, the CCE-regret is defined as*

$$\text{Regret}_{\text{CCE}}(K) = \sum_{k=1}^{K} \max_{i \in [n]} (V_i^{\dagger, \pi_{-i}^k} - V_i^{\pi^k}).$$

Finally, a correlated equilibrium is defined as a joint (potentially correlated) policy where no player can increase her value by unilaterally applying any strategy modification. Formally,

**Definition 5** (Correlated Equilibrium). *A joint policy $\pi$ is a **CE** if $\max_{\phi_i} V_i^{(\phi_i \diamond \pi_i) \odot \pi_{-i}} = V_i^{\pi}$ for all $i \in [n]$. A joint policy $\pi$ is an $\epsilon$-approximate CE if $\max_{\phi_i} V_i^{(\phi_i \diamond \pi_i) \odot \pi_{-i}} \leq V_i^{\pi} + \epsilon$ for all $i \in [m]$.*

In Partially Observable Markov games, we always have that a Nash equilibrium is a CE, and a CE is a CCE. Finally, we define the CE-regret to be the cumulative violation of the CE condition.

**Definition 6** (CE-regret). *Let $\pi^k$ denote the policy deployed by an algorithm in the $k^{th}$ episode. After a total of $K$ episodes, the CE-regret is defined as*

$$\text{Regret}_{\text{CE}}(K) = \sum_{k=1}^{K} \max_{i \in [n]} \max_{\phi_i} (V_i^{(\phi_i \diamond \pi_i^k) \odot \pi_{-i}^k} - V_i^{\pi^k}).$$

## 3 Weakly Revealing Partially Observable Markov Games

In this section, we define the class of weakly revealing POMGs. To begin with, we consider undercomplete POMGs where there are more observations than hidden states, i.e., $O \geq S$. Formally, the family of $\alpha$-weakly revealing POMGs includes all POMGs, in which the $S^{\text{th}}$ singular value of each emission matrix $\mathbb{O}_h$ is lower bounded by $\alpha > 0$.

---

**Algorithm 1** OMLE-Equilibrium

1: **Initialize:** $\mathcal{B}^1 = \{\hat{\theta} \in \Theta : \min_h \sigma_S(\hat{\mathbb{O}}_h) \geq \alpha\}, \mathcal{D} = \{\}$
2: **for** $k = 1, \ldots, K$ **do**
3:     compute $\pi^k =$ Optimistic_Equilibrium$(\mathcal{B}^k)$
4:     follow $\pi^k$ to collect a trajectory $\tau^k = (\mathbf{o}_1^k, \mathbf{a}_1^k, \ldots, \mathbf{o}_H^k, \mathbf{a}_H^k)$
5:     add $(\pi^k, \tau^k)$ into $\mathcal{D}$ and update
$$\mathcal{B}^{k+1} = \left\{\hat{\theta} \in \Theta : \sum_{(\pi,\tau) \in \mathcal{D}} \log \mathbb{P}_{\hat{\theta}}^{\pi}(\tau) \geq \max_{\theta' \in \Theta} \sum_{(\pi,\tau) \in \mathcal{D}} \log \mathbb{P}_{\theta'}^{\pi}(\tau) - \beta\right\} \bigcap \mathcal{B}^1$$
6: output $\pi^{\text{out}}$ that is sampled uniformly at random from $\{\pi^k\}_{k \in [K]}$

---

**Assumption 1** ($\alpha$-weakly revealing condition). *There exists $\alpha > 0$, such that $\min_h \sigma_S(\mathbb{O}_h) \geq \alpha$.*

Assumption 1 is simply a robust version of the condition that the rank of each emission matrix is $S$, which guarantees that no two different latent state mixtures can generate the same observation distribution, i.e., $\mathbb{O}_h \nu_1 \neq \mathbb{O}_h \nu_2$ for any different $\nu_1, \nu_2 \in \Delta_S$. Intuitively, this guarantees that the **joint observations** of all agents contain sufficient information to distinguish any two different state mixtures. We remark that this is much weaker than requiring the individual observations of each agent contain sufficient information about the latent states. The weakly revealing condition is important in excluding those pathological POMGs where the observations contain no useful information for identifying the key parts of model dynamics.

Note that Assumption 1 never holds in the overcomplete setting ($S > O$) as it is impossible to distinguish any two latent state mixtures by only inspecting the observation distribution in a single step. To address this issue, we can instead inspect the observations for $m$-consecutive steps. To proceed, we define the $m$-step emission-action matrices

$$\{\mathbb{M}_h \in \mathbb{R}^{(A^{m-1}O^m) \times S}\}_{h \in [H-m+1]}$$

as follows: Given an observation sequence $\bar{\mathbf{o}}$ of length $m$, initial state $s$ and action sequence $\bar{\mathbf{a}}$ of length $m - 1$, we let $[\mathbb{M}_h]_{(\bar{\mathbf{a}}, \bar{\mathbf{o}}), s}$ be the probability of receiving $\bar{\mathbf{o}}$ provided that the action sequence $\bar{\mathbf{a}}$ is used from state $s$ and step $h$:

$$[\mathbb{M}_h]_{(\bar{\mathbf{a}}, \bar{\mathbf{o}}), s} = \mathbb{P}(o_{h:h+m-1} = \bar{\mathbf{o}} \mid s_h = s, a_{h:h+m-2} = \bar{\mathbf{a}}), \quad \forall (\bar{\mathbf{a}}, \bar{\mathbf{o}}, s) \in \mathcal{A}^{m-1} \times \mathcal{O}^m \times \mathcal{S}. \quad (2)$$

Similar to the undercomplete setting, the weakly-revealing condition in the over-complete setting simply assumes the $S^{\text{th}}$ singular value of each $m$-step emission-action matrix is lower bounded.

**Assumption 2** (multistep $\alpha$-weakly revealing condition). *There exists $m \in \mathbb{N}$, $\alpha > 0$ such that $\min_h \sigma_S(\mathbb{M}_h) \geq \alpha$ where $\mathbb{M}_h$ is the $m$-step emission matrix defined in (2).*

Assumption 2 ensures that $m$-step consecutive observations shall contain sufficient information to distinguish any two different latent state mixtures. Note that Assumption 1 is a special case of Assumption 2 with $m = 1$. Finally, we remark that the single-agent versions of Assumption 1 and 2 were first identified in [19] and [26] as sufficient conditions for sample-efficient learning of single-step and multi-step weakly revealing POMDPs (the single-agent version of POMGs), respectively.

## 4 Learning Equilibria with Self-play

In this section, we study the self-play setting where the algorithm can control all the players to learn the equilibria by playing against itself. We propose a new algorithm — *Optimistic Maximum Likelihood Estimation for Learning Equilibria* (OMLE-Equilibrium) that can provably find Nash equilibria, coarse correlated equilibria and correlate equilibria in any weakly-revealing partially observable Markov games using a number of samples polynomial in all relevant parameters.

### 4.1 Undercomplete partially observable Markov games

We first present the algorithm and results for learning undercomplete POMGs under Assumption 1. We will see in the later section that with a minor modification the same algorithm also applies to learning overcomplete POMGs under Assumption 2.

---
**Subroutine 1** Optimistic_Equilibrium($\mathcal{B}$)
---
1: **for** $i \in [n]$ **do**
2:     let $\overline{V}_i \in \mathbb{R}^{|\Pi_1^{\text{det}}| \times \cdots \times |\Pi_n^{\text{det}}|}$ with its $\pi^{\text{th}}$ entry equal to $\sup_{\hat{\theta} \in \mathcal{B}} V_i^{\pi}(\hat{\theta})$ for $\pi \in \Pi_1 \times \cdots \times \Pi_n$
3: **return** EQUILIBRIUM($\overline{V}_1, \ldots, \overline{V}_n$)
---

**Algorithm description** To condense notations, we will use $\theta = (\mathbb{T}, \mathbb{O}, \mu_1)$ to denote the parameters of a POMG. Given a policy $\pi$ and a trajectory $\tau$, we denote by $V_i^{\pi}(\theta)$ the $i^{\text{th}}$ player's value and by $\mathbb{P}_{\theta}^{\pi}(\tau)$ the probability of observing trajectory $\tau$, both under policy $\pi$ in the POMG model parameterized by $\theta$. We describe OMLE-Equilibrium in Algorithm 1. In each episode, the algorithm executes the following two key steps:

- **Optimistic equilibrium computation** (Line 3) We first invoke Optimistic_Equilibrium (Subroutine 1) with confidence set $\mathcal{B}^k$ to compute a joint (potentially correlated) policy $\pi^k$. Formally, subroutine Optimistic_Equilibrium($\mathcal{B}^k$) consists of two components:
  - **Optimisic value estimation** (Line 1-2 of Subroutine 1) For each player $i \in [n]$ and deterministic joint policy $\pi \in \Pi_1^{\text{det}} \times \cdots \times \Pi_n^{\text{det}}$, we compute an upper bound $\overline{V}_i^{\pi}$ for the $i^{\text{th}}$ player's value under policy $\pi$ by using the most optimistic POMG model in the confidence set $\mathcal{B}^k$.
  - **Equilibira computation** (Line 3 of Subroutine 1) Given the optimistic value estimates for all deterministic joint policies and all players, we can view the POMG as a normal-form game where the $i^{\text{th}}$ player's pure strategies consist of all her deterministic policies (i.e., $\Pi_i^{\text{det}}$) and the payoff she receives under a joint deterministic policy $\pi \in \Pi_1^{\text{det}} \times \cdots \times \Pi_n^{\text{det}}$ is equal to the corresponding optimistic value estimate $\overline{V}_i^{\pi}$. Then we compute a EQUILIBRIUM $\pi^k$ for this normal-form game, which is a mixture of all the deterministic joint policies in $\Pi_1^{\text{det}} \times \cdots \times \Pi_n^{\text{det}}$.
- **Confidence set update** (Line 4-5) We first follow $\pi^k$ to collect a trajectory, and then utilize the newly collected data to update the model confidence set via MLE principle.

Here we highlight two algorithmic designs in OMLE-Equilibrium: the flexibility of equilibrium computation and the MLE confidence set construction. In the step of equilibrium computation, we can choose EQUILIBRIUM to be Nash equilibrium or correlated equilibrium (CE) or corase correlated equilibrium (CCE) of the normal-form game depending on the target type of equilibrium we aim to learn for the POMG. With regard to the confidence set design, we adopt the idea from [26] to include all the POMG models whose likelihood on the historical data is close to the maximum likelihood. This can be viewed as a relaxation of the classic MLE method, with the degree of relaxation controlled by parameter $\beta$. One important benefit of this relaxation is that although the groundtruth POMG model is in general not a solution of MLE, its likelihood ratio is rather close to the maximal likelihood. By doing so, we can guarantee the true model is included in the confidence set with high probability. Finally, we remark that Algorithm 1 is computationally inefficient in general due to the steps of optimistic value estimation and equilibrium computation.

**Theoretical guarantees** Below we present the main theorem for OMLE-Equilibrium.

**Theorem 7.** (Regret of OMLE-Equilibrium) *Under Assumption 1, there exists an absolute constant $c$ such that for any $\delta \in (0,1]$ and $K \in \mathbb{N}$, Algorithm 1 with $\beta = c\left((S^2 A + SO)\log(SAOHK) + \log(K/\delta)\right)$ and EQUILIBRIUM being one of {Nash, CCE, CE} satisfies (respectively) that with probability at least $1 - \delta$,*

$$\text{Regret}_{\{\text{Nash,CCE,CE}\}}(k) \le \text{poly}(S, A, O, H, \alpha^{-1}, \log(K\delta^{-1})) \cdot \sqrt{k} \qquad \text{for all } k \in [K].$$

Theorem 7 claims that if all players follow OMLE-Equilibrium, then the cumulative {Nash,CCE,CE}-regret is upper bounded by $\tilde{\mathcal{O}}(\sqrt{k})$ for any weakly-revealing POMGs that satisfy Assumption 1, where the growth rate w.r.t $k$ is optimal. By the standard online-to-batch conversion, it directly implies the following sample complexity result:

**Corollary 8.** (Sample Complexity of OMLE-Equilibrium) *Under the same setting as Theorem 7, when $K \ge \text{poly}(S, A, O, H, \alpha^{-1}, \log(\epsilon^{-1}\delta^{-1})) \cdot \epsilon^{-2}$, then with probability at least $1/2$, $\pi^{\text{out}}$ is an $\epsilon$-{Nash, CCE, CE} policy.*

---

**Algorithm 2** multi-step OMLE-Equilibrium

---

1: **Initialize:** $\mathcal{B}^1 = \{\hat{\theta} \in \Theta : \min_h \sigma_S(\hat{\mathbb{M}}_h) \geq \alpha\}$, $\mathcal{D} = \{\}$
2: **for** $k = 1, \ldots, K$ **do**
3:     compute $\pi^k$ =Optimistic_Equilibrium$(\mathcal{B}^k)$
4:     **for** $h = 0, \ldots, H - m$ **do**
5:         execute policy $\pi^k_{1:h} \circ \text{uniform}(\mathcal{A})$ to collect a trajectory $\tau^{k,h}$
           then add $(\pi^k_{1:h} \circ \text{uniform}(\mathcal{A}), \tau^{k,h})$ into $\mathcal{D}$
6:     update
$$\mathcal{B}^{k+1} = \left\{ \hat{\theta} \in \Theta : \sum_{(\pi,\tau) \in \mathcal{D}} \log \mathbb{P}^{\pi}_{\hat{\theta}}(\tau) \geq \max_{\theta' \in \Theta} \sum_{(\pi,\tau) \in \mathcal{D}} \log \mathbb{P}^{\pi}_{\theta'}(\tau) - \beta \right\} \bigcap \mathcal{B}^1$$

7: output $\pi^{\text{out}}$ that is sampled uniformly at random from $\{\pi^k\}_{k \in [K]}$

---

Here the dependence on the precision parameter $\epsilon$ is optimal. Finally, notice that the upper bound in Theorem 7 depends polynomially on the inverse of $\alpha$ — a lower bound for the minimal singular value of the joint emission matrix $\mathbb{O}_h$ in Assumption 1. This dependence is shown to be unavoidable even in the single-player setting (POMDPs) [26].

## 4.2 Overcomplete partially observable Markov games

In this subsection, we extend OMLE-Equilibrium to the more challenging setting of learning overcomplete POMGs, where there can be much less observations than latent states. We prove that a simple variant of OMLE-Equilibrium still enjoys polynomial sample-efficiency guarantee for learning any multi-step weakly revealing POMGs.

**Algorithm description**   We describe the multi-step generalization of OMLE-Equilibrium in Algorithm 2, which inherits the key designs from Algorithm 1 and additionally makes two important modifications to address the challenge of insufficient information from single-step observation. The first change is to utilize a more active sampling strategy for exploration. Instead of simply following the optimistic policy $\pi^k$, we will iteratively execute $H - m + 1$ policies of form $\pi^k_{1:h} \circ \text{uniform}(\mathfrak{A})$ where the players first follow policy $\pi^k$ from step 1 to step $h$, then pick actions uniformly at random to finish the remaining $H - h$ steps. Intuitively, by actively trying random action sequences after executing policy $\pi^k$, the algorithm can acquire more information about the system dynamics corresponding to those latent states that are frequently visited by $\pi^k$, and therefore help address the challenge of lacking sufficient information from single-step observation. The second change made by Algorithm 2 is that in constructing the confidence set, we require the minimal singular value of the multistep emission matrix to be lower bounded, which enforces the multistep weakly revealing condition in Assumption 2.

**Theoretical guarantee**   Below we present the main theorem for multi-step OMLE-Equilibrium.

**Theorem 9.** (Total suboptimality of multi-step OMLE-Equilibrium) *Under Assumption 2, there exists an absolute constant $c$ such that for any $\delta \in (0, 1]$ and $K \in \mathbb{N}$, Algorithm 1 with $\beta = c\left( (S^2 A + SO) \log(SAOHK) + \log(K/\delta) \right)$ and* EQUILIBRIUM *being one of {Nash, CCE, CE} satisfies (respectively) that with probability at least $1 - \delta$,*

$$\text{Regret}_{\{\text{Nash,CCE,CE}\}}(k) \leq \text{poly}(S, A^m, O, H, \alpha^{-1}, \log(K\delta^{-1})) \cdot \sqrt{k} \qquad \textit{for all } k \in [K],$$

*where the regret is computed for policy $\pi^1, \ldots, \pi^k$.*

Theorem 9 claims that the total {Nash,CCE,CE}-"regret" (that are computed on policy $\pi^1, \ldots, \pi^k$) of multi-step OMLE-Equilibrium is upper bounded by $\tilde{\mathcal{O}}(\sqrt{k})$ for any multi-step weakly revealing POMGs satisfying Assumption 2. We remark that, strictly speaking, Theorem 9 is not a standard regret guarantee since the policies executed by multi-step OMLE-Equilibrium are compositions of $\pi^1, \ldots, \pi^k$ and random actions, instead of purely $\pi^1, \ldots, \pi^k$. Nevertheless, we can still utilize the standard online-to-batch conversion to obtain the following sample complexity guarantee:

**Corollary 10.** (sample complexity of multi-step OMLE-Equilibrium) *Under the same setting as Theorem 9, when $K \geq \text{poly}(S, A^m, O, H, \alpha^{-1}, \log(\epsilon^{-1}\delta^{-1})) \cdot \epsilon^{-2}$, then with probability at least $1/2$, $\pi^{\text{out}}$ is an $\epsilon$-{Nash, CCE, CE} policy.*

**Algorithm 3** OMLE-Adversary

1: **Initialize:** $\mathcal{B}^1 = \{\hat{\theta} \in \Theta : \sigma_S(\hat{\mathbb{O}}) \geq \alpha\}$, $\mathcal{D} = \{\}$
2: **for** $k = 1, \ldots, K$ **do**
3:    learner computes $(\cdot, \pi_1^k) = \mathrm{argmax}_{\hat{\theta} \in \mathcal{B}^k, \hat{\pi}_1 \in \Pi_1} \min_{\hat{\pi}_{-1} \in \Pi_{-1}} V_1^{\hat{\pi}_1 \times \hat{\pi}_{-1}}(\hat{\theta})$
4:    opponents pick policies $\pi_{-1}^k$
5:    execute policy $\pi^k = \pi_1^k \times \pi_{-1}^k$ to collect $\tau^k = (\mathbf{o}_1^k, \mathbf{a}_1^k, \ldots, \mathbf{o}_H^k, \mathbf{a}_H^k)$
6:    add $(\pi^k, \tau^k)$ into $\mathcal{D}$ and update
$$\mathcal{B}^{k+1} = \left\{ \hat{\theta} \in \Theta : \sum_{(\pi,\tau) \in \mathcal{D}} \log \mathbb{P}_{\hat{\theta}}^\pi(\tau) \geq \max_{\theta' \in \Theta} \sum_{(\pi,\tau) \in \mathcal{D}} \log \mathbb{P}_{\theta'}^\pi(\tau) - \beta \right\} \bigcap \mathcal{B}^1 \qquad (3)$$

Here the dependence on the precision parameter $\epsilon$ is optimal up to poly-logarithmic factors. Finally, observe that the sample complexity has $A^m$ dependency that is exponential in $m$, which is unavoidable in general even in the single-player setting (POMDPs) [26]. Nonetheless, in order to make $\min_h \mathrm{rank}(\mathbb{M}_h) = S$ possible (Assumption 2), we only need to make $(OA)^m \gtrsim S$, i.e., $m \gtrsim \log S$ which is very small. In this paper, when we claim the sample complexity is polynomial, we consider $m$ to be small enough so that $A^m \leq \mathrm{poly}(S, A, O, H, \alpha^{-1})$.

## 5 Playing against Adversarial Opponents

In this section, we turn to the online setting where the learner only controls a single player and the remaining players can execute arbitrary strategies. In this setting, we no longer target at learning game-theoretic equilibria because if other players keep playing some highly suboptimal policies then the learner may never be able to explore the environment thoroughly and thus lacks sufficient information to compute equilibria. Instead, we consider the standard goal for online setting, which is to achieve low regret in terms of cumulative rewards even if all other players play adversarially against the learner. Without loss of generality, we assume the learner only controls the $1^{\mathrm{st}}$ player throughout this section.

### 5.1 Statistical hardness for the standard setting

We first consider the standard POMG setting where each player can only observe her *own* observations and actions. We prove that achieving low regret in this setting is impossible in general even if (i) the POMG is two-player zero-sum and satisfies Assumption 1 with $\alpha = 1$, (ii) the opponent keeps playing a fixed deterministic policy *known* to the learner, and (iii) the only parts of the model unknown to the learner are the emission matrices.

**Theorem 11.** *For any $L, k \in \mathbb{N}^+$, there exist (i) a two-player zero-sum POMG of size $S, A, O, H = \mathcal{O}(L)$ and satisfying Assumption 1 with $\alpha = 1$, and (ii) a fixed opponent who keeps playing a known deterministic policy $\pi_2$, so that with probability at least $1/2$*

$$\sum_{t=1}^k \left( \max_{\tilde{\pi}_1} \min_{\tilde{\pi}_2} V_1^{\tilde{\pi}_1 \times \tilde{\pi}_2} - V_1^{\pi_1^t \times \pi_2} \right) \geq \Omega \left( \min\{2^L, k\} \right),$$

*where $\pi_1^t$ is the policy played by the learner in the $t^{\mathrm{th}}$ episode.*

Theorem 11 claims that when the learner is not allowed to access the opponent's observations and actions, there exists exponential regret lower bound for competing with the max-min value (i.e., Nash value in two-player zero-sum POMGs) even in the very benign scenario as described above. We remark that this lower bound directly implies competing with the best fixed policy in hindsight is also hard because the max-min value is always no larger than the value of the best-response to $\pi_2$:

$$\max_{\tilde{\pi}_1} \min_{\tilde{\pi}_2} V_1^{\tilde{\pi}_1 \times \tilde{\pi}_2} \leq \max_{\tilde{\pi}_1} V_1^{\tilde{\pi}_1 \times \pi_2} = V_1^{\dagger, \pi_2}.$$

### 5.2 Positive results for the game-replay setting

In this section, we consider the game-replay setting where *after* each episode of play, every player will reveal their observations and actions in this episode to other players. In other words, every player

is able to observe the whole trajectory $\tau^k = (\mathbf{o}_1^k, \mathbf{a}_1^k, \ldots, \mathbf{o}_H^k, \mathbf{a}_H^k)$ *after* the $k^{\text{th}}$ episode is finished. The motivation for considering this setting is in many real-world games, e.g., Dota, StarCraft and Poker, players are usually allowed to watch the replays of the games they have played, in which they can freely view other players' observations and actions. Below, we show that a simple variant of OMLE-Equilibrium enjoys sublinear regret when playing against adversarial opponents.

**Algorithm description**   We provide the formal description of OMLE-Adversary in Algorithm 3. Same as OMLE-Equilibrium, OMLE-Adversary utilizes the relaxed MLE approach to construct the confidence set. The key modification lies in the computation of player 1's (stochastic) policy $\pi_1^k$ (Line 3). Specifically, the learner will compute the most optimistic model $\theta^k$ in the confidence set $\mathcal{B}^k$ by examining player 1's max-min value in each model. Then choose player 1's policy to be the one with the highest value under $\theta^k$, assuming all other players jointly play against player 1.

Finally, we remark that although the confidence set construction in Line 6 seems to involve the joint policies $\pi$ of all players, the confidence set itself is in fact independent of the joint policies $\pi$. Therefore, the $1^{\text{st}}$ player can still construct the confidence set without knowledge of other players' policies. This is because the dependency of the loglikelihood function on policy $\pi$ are equal on both sides of (3), and thus they cancel with each other. Formally, for any $\hat{\theta}, \theta' \in \Theta$, we have

$$\sum_{t=1}^{k} \left( \log \mathbb{P}_{\hat{\theta}}^{\pi^t}(\tau^t) - \log \mathbb{P}_{\theta'}^{\pi^t}(\tau^t) \right) = \sum_{t=1}^{k} \left( \log \mathbb{P}_{\hat{\theta}}(\mathbf{o}_{1:H}^t \mid \mathbf{a}_{1:H}^t) - \log \mathbb{P}_{\theta'}(\mathbf{o}_{1:H}^t \mid \mathbf{a}_{1:H}^t) \right).$$

**Theoretical guarantees**   Below we present the main theorem for OMLE-Adversary.

**Theorem 12.** (Regret of OMLE-Adversary) *Under Assumption 1, there exists an absolute constant $c$ such that for any $\delta \in (0, 1]$ and $K \in \mathbb{N}$, Algorithm 3 with $\beta = c\left( (S^2 A + SO) \log(SAOHK) + \log(K/\delta) \right)$ satisfies that with probability at least $1 - \delta$,*

$$\sum_{t=1}^{k} \left( \max_{\tilde{\pi}_1} \min_{\tilde{\pi}_2} V_1^{\tilde{\pi}_1 \times \tilde{\pi}_2} - V_1^{\pi^t} \right) \le \mathrm{poly}(S, A, O, H, \alpha^{-1}, \log(K\delta^{-1})) \cdot \sqrt{k} \quad \text{for all } k \in [K].$$

Theorem 12 claims that the regret of OMLE-Adversary is upper bounded by $\tilde{\mathcal{O}}(\sqrt{k})$ in any weakly revealing POMGs that satisfy Assumption 1, no matter what adversarial strategies other players might take. Here the regret is defined by comparing the cumulative rewards received by player 1 to the max-min value $\max_{\tilde{\pi}_1} \min_{\tilde{\pi}_2} V_1^{\tilde{\pi}_1 \times \tilde{\pi}_2}$ that is the largest value she could receive if all other players jointly play against her. Notice that this regret is weaker than the typical version of regret considered in online learning literature, which typically competes with the best response in hindsight, i.e.,

$$\max_{\tilde{\pi}_1} \sum_{t=1}^{k} \left( V_1^{\tilde{\pi}_1 \times \pi_{-1}^t} - V_1^{\pi^t} \right).$$

Therefore, it is natural to ask whether we can obtain similar sublinear regret in terms of the above regret definition. Unfortunately, previous work [27] proved that there exists exponential regret lower bound for competing with the best response in hindsight even in *fully observable* two-player zero-sum Markov games, which are special cases of POMGs satisfying Assumption 1 with $\alpha = 1$. As a result, achieving low regret in the above sense is also intractable in POMGs.

**Generalization to multi-step weakly revealing POMGs**   So far, we only derive the positve result (Theorem 12) for single-step weakly revealing POMGs. A reader might wonder whether similar results can be obtained in the more general setting of multi-step weakly revealing POMGs. Unfortunately, this generalization turns out to be impossible in general, even if (i) the POMG satisfies Assumption 2 with $m = 2$ and $\alpha \ge 1$, and (ii) the learner can directly observe the opponents' actions and observations. We defer the formal statement of this hardness result and its proof to Appendix D.3.

## Acknowledgements

Csaba Szepesvári gratefully acknowledges the funding from Natural Sciences and Engineering Research Council (NSERC) of Canada, "Design.R AI-assisted CPS Design" (DARPA) project and the Canada CIFAR AI Chairs Program for Amii. Chi Jin gratefully acknowledges the Project X innovation fund from Princeton.

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
