# A   On Relation between IIEFGs and Weakly Revealing POMGs

In this section, we consider imperfect-information extensive-form games with perfect recall, which we call IIEFGs for simplicity. In this section, we will show that any IIEFG (of a polynomial size) is also a weakly revealing POMG (of a polynomial size) that satisfies Assumption 1 with $\alpha = 1$. As a result, all the algorithms and the polynomial sample complexity results developed in this paper immediately apply to learning IIEFGs using polynomial samples.

We note that the reverse is not true, we can easily construct a weakly-revealing POMGs of a polynomial size that can not be represented by any IIEFGs with a polynomial size, due to the restriction of tree-structured transition and deterministic transition in IIEFGs. Therefore, polynomial sample complexity for learning IIEFGs does not imply polynomial sample complexity results for learning POMGs.

## A.1   Representing IIEFGs as 1-Weakly Revealing POMGs

We first introduce the definition of IIEFGs. There are many equivalent formulations of IIEFGs and here we adopt the formulation used in [24], which allows a clearer comparison to POMGs.

**Definition 13.** *An* imperfect information extensive-form game with perfect recall[4] *is a POMG* $(H, \mathcal{S}, \{\mathcal{A}_i\}_{i=1}^n, \{\mathcal{O}_i\}_{i=1}^n; \mathbb{T}, \mathbb{O}, \mu_1; \{r_i\}_{i=1}^n)$ *that additionally satisfies the followings:*

- *Tree-structured transition: for each $s \in \mathcal{S}$ and $h \in [H-1]$, there is at most one state-action pair $(s', \mathbf{a}') \in \mathcal{S} \times \mathcal{A}$ such that $\mathbb{T}_h(s \mid s', \mathbf{a}') \neq 0$. In other words, for any $s_h$, there is a unique history sequence $(s_1, \mathbf{a}_1, \ldots, s_{h-1}, \mathbf{a}_{h-1})$ that leads to $s_h$.*

- *Deterministic emission and perfect-recall: for each $s \in \mathcal{S}$ and $h \in [H]$, $\|\mathbb{O}_h(\cdot \mid s)\|_0 = 1$. That is, no state can emit two different observations. Moreover, for each player $i$ and $x \in \mathcal{O}_i$, there is a unique history $(o_{i,1}, a_{i,1}, \ldots, o_{i,h} = x)$ up to $x$ from player $i$'s perspective. This means player $i$ can always retrieve her previous observations and actions solely from her current-step observation. In IIEFGs, the observations are usually referred to as* information sets.

- *Delayed and state-action-dependent reward: different from our definition of reward in Section 2, now each $r_{i,h}$ is a random function from $\mathcal{S} \times \mathcal{A}$ to $[0,1]$, and the rewards are revealed to each learner only at the end of each episode. In other words, player $i$ gets to observe $r_{i,1}^k, \ldots, r_{i,H}^k$ after the $k^{\text{th}}$ episode is finished.* [5]

We now show that any IIEFG can be represented by 1-weakly-revealing POMGs.

**Theorem 14.** *Any IIEFG$(H, \mathcal{S}, \{\mathcal{A}_i\}_{i=1}^n, \{\mathcal{O}_i\}_{i=1}^n; \mathbb{T}, \mathbb{O}, \mu_1; \{r_i\}_{i=1}^n)$ can be represented as a POMG with $\prod_i |\mathcal{O}_i|$ states, the same action space, the same observation space, stochastic rewards which depend on the joint observation and action, and satisfying the single-step weakly revealing condition (Assumption 1) with $\alpha = 1$.*

Theorem 14 shows that any IIEFG of a polynomial size can be efficiently represented by 1-weakly-revealing POMGs with a polynomial size. Here, we consider the number of player $n$ as constant when discussing polynomial versus exponential.

*Proof of Theorem 14.* We consider an equivalent POMG formulation denoted as

$$(H, \tilde{\mathcal{S}}, \{\mathcal{A}_i\}_{i=1}^n, \{\mathcal{O}_i\}_{i=1}^n; \tilde{\mathbb{T}}, \tilde{\mathbb{O}}, \tilde{\mu}_1; \{\tilde{r}_i\}_{i=1}^n)$$

where we highlight the modified parts in blue and define them as following:

- **State and transition.** Notice that the joint observation in IIEFGs always satisfies the Markov property because of the perfect-recall emission structure:

$$\mathbb{P}(\mathbf{o}_{h+1} \mid \mathbf{o}_1, \mathbf{a}_1, \ldots, \mathbf{o}_h, \mathbf{a}_h) = \mathbb{P}(\mathbf{o}_{h+1} \mid \mathbf{o}_h, \mathbf{a}_h).$$

---

[4]Strictly speaking, we restrict our attention to timeable IIEFGs. We remark that, as argued by [16], non-timeable IIEFGs can not be implemented in practical systems.

[5]We WLOG consider the delayed reward since almost all algorithms in the IIEFG literature only use information sets (i.e., do not use additional information in the intermediate reward) to make decisions.

Therefore we can view the original joint observation space as the new state space $\tilde{\mathcal{S}} := \prod_i \mathcal{O}_i$ and define the transition as

$$\tilde{\mathbb{T}}_h(\tilde{s}' \mid \tilde{s}, \mathbf{a}) := \mathbb{P}(\mathbf{o}_{h+1} = \tilde{s}' \mid \mathbf{o}_h = \tilde{s}, \mathbf{a}_h = \mathbf{a}).$$

And the initial distribution is defined as $\tilde{\mu}_1 := \mathbb{P}(\mathbf{o}_1 = \cdot \mid s_1 \sim \mu_1)$.

- **Emission.** We define the emission so that player $i$ always observes $[\tilde{s}_h]_i$ (the $i^{\text{th}}$ entry in $\tilde{s}_h$) with probability 1 at step $h$. Formally for all $h \in [H]$ and $(\mathbf{o}, \tilde{s}) \in \mathcal{O} \times \tilde{\mathcal{S}}$

$$\tilde{\mathbb{O}}_h(\mathbf{o} \mid \tilde{s}) = \mathbf{1}(\mathbf{o} = \tilde{s}).$$

Clearly, in this case the joint emission is identity and therefore satisfies the single-step weakly revealing condition (Assumption 1) with $\alpha = 1$.

- **Reward.** As for the reward function, we let $\tilde{r}_{i,h} := 0$ for $h \leq H - 1$, and define $\tilde{r}_{i,H}(\mathbf{o}_H, \mathbf{a}_H)$ to be a random variable taking value $\sum_{h=1}^{H} r_{i,h}(s_h, \mathbf{a}_h)$ with $s_{1:H}$ sampled from

$$\mathbb{P}(s_{1:H} = \cdot \mid \mathbf{o}_1, \mathbf{a}_1, \ldots, \mathbf{o}_H, \mathbf{a}_H) = \mathbb{P}(s_{1:H} = \cdot \mid \mathbf{o}_H, \mathbf{a}_H).$$

Therefore, the reward $\tilde{r}_{i,H}$ is a random function of the joint observation and action $(\mathbf{o}_H, \mathbf{a}_H)$ at step $H$.

It is direct to see any policy induces the same distribution over $\mathbf{o}_1, \mathbf{a}_1, \ldots, \mathbf{o}_H, \mathbf{a}_H$ and enjoys the same value in this new formulation as in the original IIEFG. As a result, any algorithms designed for weakly revealing POMGs also apply to learning IIEFGs.

**Stochastic reward depending on the joint observation and action**    Recall when defining POMGs in Section 2, we let the reward to be a deterministic function of individual observations. Nonetheless, one can easily verify all our results in this paper still hold without non-trivial modifications when the reward functions are stochastic and depend on the joint observation and action. As a result, we conclude that any IIEFG with $O$ observations, $S$ latent states and $A$ actions can be represented as a 1-weakly revealing POMG with $O$ observations, $O$ latent states and $A$ actions, to which all our algorithms and theoretical guarantees directly apply.    $\square$

**Regarding the curse of multi-player**    Note that all the sample complexity results proved in this paper scale exponentially with respect to $n$, the number of players. Therefore, they suffer from the curse of multi-players when $n$ is large. In particular, when specializing these results to the setting of IIEFGs, we obtain sample complexity scaling with $\prod_{i \in [n]} |\mathcal{O}_i|$ instead of $\sum_{i \in [n]} |\mathcal{O}_i|$ where the latter is achievable by algorithms specially designed for learning IIEFGs [e.g., 4]. Nonetheless, we observe that the 1-weakly revealing POMG presentation of IIEFGs derived in Theorem 14 possesses additional benign structures: the state space is factored and the emission is identity, which could potentially be exploited to overcome the curse of dimensionality with sharper analysis or different algorithm design (e.g., incorporate the idea of V-learning style algorithms [21, 38, 11]).

## A.2    On Inefficiency of Representing POMGs using IIEFGs

We prove two theorems for representing POMGs using IIEFGs:

- First we show POMGs can be represented by IIEFGs with an exponentially large size. (Exponential large model is prohibitive in practice, more relevant question is for the polynomial size).

- Then we prove a lower bound showing that there exists weakly revealing POMGs of constant size, which can not be represented by any IIEFGs with a polynomial size. This implies IIEFGs can not efficiently represent POMGs and polynomial results for learning IIEFGs can not translate into efficiency guarantees for learning POMGs.

**Theorem 15.** *A POMG$(H, \mathcal{S}, \{\mathcal{A}_i\}_{i=1}^{n}, \{\mathcal{O}_i\}_{i=1}^{n}; \mathbb{T}, \mathbb{O}, \mu_1; \{r_i\}_{i=1}^{n})$ can be represented as an IIEFG with $(\prod_i |\mathcal{O}_i||\mathcal{A}_i|)^H$ states, the same action space, $(|\mathcal{O}_i||\mathcal{A}_i|)^H$ observations for each player $i \in [n]$.*

*Proof of Theorem 15.* We consider an equivalent IIEFG formulation denoted as

$$(H, \tilde{\mathcal{S}}, \{\mathcal{A}_i\}_{i=1}^n, \{\mathcal{O}_i\}_{i=1}^n; \tilde{\mathbb{T}}, \tilde{\mathbb{O}}, \tilde{\mu}_1; \{\tilde{r}_i\}_{i=1}^n)$$

where we highlight the modified parts in blue and define them as following:

- **State and transition.** We view the entire interaction history as the state of IIEFG, that is, $\tilde{s}_h = (\mathbf{o}_1, \mathbf{a}_1, \ldots, \mathbf{o}_h)$. Under such choice of latent state, the transition is clearly tree structured and satisfies: for any $\tilde{s}_h = (\mathbf{o}_1, \mathbf{a}_1, \ldots, \mathbf{o}_h)$ and $\tilde{s}_{h+1} = (\mathbf{o}'_1, \mathbf{a}'_1, \ldots, \mathbf{o}'_{h+1})$

$$\tilde{\mathbb{T}}_h(\tilde{s}_{h+1} \mid \tilde{s}_h, \mathbf{a}_h) = \mathbb{P}(\mathbf{o}'_{h+1} \mid (\mathbf{o}', \mathbf{a}')_{1:h}) \times \mathbf{1}((\mathbf{o}, \mathbf{a})_{1:h} = (\mathbf{o}', \mathbf{a}')_{1:h}).$$

- **Emission.** We define the emission so that player $i$ always observes $[\tilde{s}_h]_i$ (the $i^{\text{th}}$ entry in $\tilde{s}_h$) with probability 1 at step $h$. Formally for all $h \in [H]$, if the environment is at state $\tilde{s}_h = (\mathbf{o}_1, \mathbf{a}_1, \ldots, \mathbf{o}_h)$, then each player $i$ will observe $(o_{i,1}, a_{i,1}, \ldots, o_{i,h})$ with probability 1. By the definition of state and transition, $[\tilde{s}_h]_i$ is exactly equal to the interaction history of player $i$. Therefore, such emission structure satisfies the perfect-recall condition.

- **Reward.** As for the reward function, we let $\tilde{r}_{i,h} := 0$ for $h \leq H - 1$. At step $H$, for any $\tilde{s}_H = (\mathbf{o}_1, \mathbf{a}_1, \ldots, \mathbf{o}_H)$ and $\mathbf{a}_H$, we define

$$\tilde{r}_{i,H}(\tilde{s}_H, \mathbf{a}_H) = \sum_{h=1}^H r_{i,h}(o_{i,h}, a_{i,h}).$$

$\square$

**Theorem 16.** *There exists an $\Omega(1)$-weakly revealing POMG of size $\mathcal{O}(1)$, which is not equivalent to any perfect-recall IIEFG with $\max_{i \in [n]} |\mathcal{O}_i| \leq 4^{H-1}$.*

*Proof of Theorem 16.* It suffices to prove the above theorem for the single-agent case, i.e., POMDPs. Consider a POMDP with 2 states, 2 actions and 2 observations. The emission and transition is defined as

$$\mathbb{T}_{h,i} = \begin{pmatrix} \alpha_{h,i} & 1 - \alpha_{h,i} \\ 1 - \alpha_{h,i} & \alpha_{h,i} \end{pmatrix} \quad \text{and} \quad \mathbb{O}_h = \begin{pmatrix} \beta_h & 1 - \beta_h \\ 1 - \beta_h & \beta_h \end{pmatrix},$$

where $\{\alpha_{h,i}\}_{(h,i) \in [H] \times [2]}$ and $\{\beta_h\}_{h \in [H]}$ are i.i.d. sampled from $[0, 1/2]$. To represent the above POMDP as IIEFG with perfect recall, the size of the observation space must be at least $4^{H-1}$ since there are $4^{H-1}$ different possible trajectories of form $(o_1, a_1, \ldots, o_{H-1}, a_{H-1})$. $\square$

# B Notations

We first introduce some notations that will be frequently used in the remainder of appendix.

- We will use $\mu \in \Pi^{\text{det}}$ to refer to a *deterministic* joint policy, and use $\mu_i \in \Pi_i^{\text{det}}$ to refer to a *deterministic* policy of player $i$.
- Since each stochastic joint policy $\pi \in \Pi$ is equivalent to a distribution over all the deterministic joint polices $\Pi^{\text{det}}$, with slight abuse of notation, we denote by $\mu \sim \pi$ the process of sampling a deterministic joint policy $\mu$ from the policy distribution specified by $\pi$. We can similarly define $\mu_i \sim \pi_i$ for any stochastic policy $\pi_{=i}$ of player $i$.
- Given a policy $\pi$ and a POMG model $\theta$, denote by $\mathbb{P}_\theta^\pi$ the distribution over trajectories (i.e., $\tau_H$) produced by executing policy $\pi$ in a POMG parameterized by $\theta$. Since the reward per trajectory is bounded by $H$, we always have

$$V_i^\pi(\theta) - V_i^\pi(\hat{\theta}) \leq H \|\mathbb{P}_\theta^\pi - \mathbb{P}_{\hat{\theta}}^\pi\|$$

for any policy $\pi$, POMG models $\theta, \hat{\theta}$, and player $i$.

- Denote by $\theta^\star$ the parameters of the groundtruth POMG model we are interacting with.

# C  Proofs for the Self-play Setting

## C.1  Proof of Theorem 7

In this section, we prove Theorem 7 with a specific polynomial dependency as stated in the following theorem.

**Theorem 17.** (Regret of OMLE-Equilibrium) *Under Assumption 1, there exists an absolute constant $c$ such that for any $\delta \in (0,1]$ and $K \in \mathbb{N}$, Algorithm 1 with $\beta = c\left(H(S^2 A + SO)\log(SAOHK) + \log(K/\delta)\right)$ and* EQUILIBRIUM *being one of {Nash, CCE, CE} satisfies (respectively) that with probability at least $1 - \delta$,*

$$\text{Regret}_{\{\text{Nash,CCE,CE}\}}(k) \leq \tilde{\mathcal{O}}\left(\frac{S^2 AO}{\alpha^2}\sqrt{k(S^2 A + SO)} \times \text{poly}(H)\right) \qquad \text{for all } k \in [K].$$

The proof consists of three steps:

1. First we rewrite Algorithm 1 in an equivalent form that is perfectly compatible with the analysis in [26].

2. After that we can directly import the theoretical guarantees from [26] and obtain a sublinear upper bound for the cumulative error of density estimation.

3. Finally, we combine the game-theoretic analysis tailored for POMGs with the density estimation guarantee derived in the second step, which gives the desired sublinear game-theoretic regret.

### C.1.1  Step 1

To begin with, we make the following observations about Algorithm 1:

- The sampling procedure in each episode $k$ is equivalent to: first sample a *deterministic* joint policy $\mu^k$ from $\pi^k$ and then execute $\mu^k$ to collect a trajectory $\tau^k$.

- In constructing the confidence set $\mathcal{B}^k$, we can replace $\pi^k$ with $\mu^k$ without making any difference, because the dependency of the log-likelihood function on policy $\pi$ are equal on both sides of the inequality in $\mathcal{B}^k$ and thus they cancel with each other. Formally, for any $\hat{\theta}, \theta' \in \Theta$, we have

$$\sum_{t=1}^{k}\left(\log \mathbb{P}_{\hat{\theta}}^{\pi^t}(\tau^t) - \log \mathbb{P}_{\theta'}^{\pi^t}(\tau^t)\right)$$

$$= \sum_{t=1}^{k}\left(\log \mathbb{P}_{\hat{\theta}}(\mathbf{o}_{1:H}^t \mid \mathbf{a}_{1:H}^t) - \log \mathbb{P}_{\theta'}(\mathbf{o}_{1:H}^t \mid \mathbf{a}_{1:H}^t)\right) = \sum_{t=1}^{k}\left(\log \mathbb{P}_{\hat{\theta}}^{\mu^t}(\tau^t) - \log \mathbb{P}_{\theta'}^{\mu^t}(\tau^t)\right).$$

Based on the above two observations, Algorithm 1 can be *equivalently* written in the form of Algorithm 4 where we highlight the modified parts in blue.

**Remark 18.** *The technical reason for rewriting Algorithm 1 in the form of Algorithm 4 is that in the optimistic equilibrium subroutine (Subroutine 1) we utilize the optimistic value estimate for each* deterministic *joint policy to construct the optimistic normal-form game and compute the optimistic game-theoretic equilibria. As a result, in order to control the cumulative regret due to over-optimism, we need guarantees on the accuracy of optimistic value estimates for* deterministic *joint policies. This is why we want to explicitly insert the "dummy" deterministic policy $\mu^k$ in each episode.*

### C.1.2  Step 2

Now we can directly instantiate the analysis of optimistic MLE (Appendix E in [26]) on Algorithm 4, which gives the following theoretical guarantee:

**Theorem 19.** ([26]) *Under Assumption 1 and the same choice of $\beta$ as in Theorem 7, with probability at least $1 - \delta$, Algorithm 4 satisfies that for **all** $k \in [K]$ and **all** $\theta^1 \in \mathcal{B}^1, \ldots, \theta^K \in \mathcal{B}^K$*

- $\theta^\star \in \mathcal{B}^k$ ,

**Algorithm 4** OMLE-Equilibrium

1: **Initialize:** $\mathcal{B}^1 = \{\hat{\theta} \in \Theta : \min_h \sigma_S(\hat{\mathbb{O}}_h) \geq \alpha\}, \mathcal{D} = \{\}$
2: **for** $k = 1, \ldots, K$ **do**
3:     compute $\pi^k =$Optimistic_Equilibrium$(\mathcal{B}^k)$
4:     sample a deterministic joint plicy $\mu^k$ from $\pi^k$, then follow $\mu^k$ to collect a trajectory $\tau^k$
5:     add $(\mu^k, \tau^k)$ into $\mathcal{D}$ and update

$$\mathcal{B}^{k+1} = \left\{ \hat{\theta} \in \Theta : \sum_{(\pi,\tau) \in \mathcal{D}} \log \mathbb{P}^{\pi}_{\hat{\theta}}(\tau) \geq \max_{\theta' \in \Theta} \sum_{(\pi,\tau) \in \mathcal{D}} \log \mathbb{P}^{\pi}_{\theta'}(\tau) - \beta \right\} \bigcap \mathcal{B}^1$$

- $\sum_{t=1}^{k} \|\mathbb{P}^{\mu^t}_{\theta^t} - \mathbb{P}^{\mu^t}_{\theta^\star}\|_1 \leq \tilde{\mathcal{O}}\left( \frac{S^2 AO}{\alpha^2} \sqrt{k(S^2 A + SO)} \times \text{poly}(H) \right).$

We comment that there are two differences between the optimistic MLE algorithm in [26] and the Algorithm 4 here: (i) the former one is designed for single-player POMGs, i.e., POMDPs while the latter one is for multi-player POMGs; (ii) $\mu^t$ is computed using different criteria. Nonetheless, we can still reuse their theoretical guarantees proved in their Appendix E without making any change because: (i) in the self-play setting, multi-player POMGs can be viewed as POMDPs with a single meta-player whose action space is of cardinality $A = A_1 \times \cdots \times A_n$ and observation space is of cardinality $O = O_1 \times \cdots \times O_n$; (ii) when proving the second statement in Theorem 19, [26] only use the fact that $\tau^t$ is sampled from $\mu^t$ but allow both $\mu^t$ and $\theta^t \in \mathcal{B}^t$ to be arbitrarily chosen. (The only place [26] need to use how $\mu^t$ and $\theta^t$ is computed is in relating the regret to $\sum_{t=1}^{k} \|\mathbb{P}^{\mu^t}_{\theta^t} - \mathbb{P}^{\mu^t}_{\theta^\star}\|_1$, which has nothing to do with the proof of Theorem 19.)

### C.1.3   Step 3

Now let us prove Theorem 17 conditioning on the two relations stated in Theorem 19 being true. To proceed, we define

$$\overline{V}_i^{k,\mu} = \max_{\hat{\theta} \in \mathcal{B}^k} V_i^{\mu}(\hat{\theta}) \quad \text{for any } (\mu, k, i) \in \Pi^{\text{det}} \times [K] \times [n].$$

Note that conditioning on the first relation in Theorem 19, we always have $\overline{V}_i^{k,\mu} \geq V_i^{\mu}$ for all $(\mu, k, i) \in \Pi^{\text{det}} \times [K] \times [n]$ because by definition $V_i^{\mu} = V_i^{\mu}(\theta^\star)$.

**Nash equilibrium**   When we choose EQUILIBRIUM in Subroutine 1 to be Nash equilibrium, by the definition of Nash-regret,

$$
\begin{aligned}
\text{Regret}_{\text{Nash}}(K) &= \sum_k \max_i \left( \max_{\mu_i \in \Pi_i^{\text{det}}} V_i^{\mu_i \times \pi^k_{-i}} - V_i^{\pi^k} \right) \\
&= \sum_k \max_i \left( \max_{\mu_i \in \Pi_i^{\text{det}}} \mathbb{E}_{\mu_{-i} \sim \pi^k_{-i}} \left[ V_i^{\mu_i \times \mu_{-i}} \right] - V_i^{\pi^k} \right) \\
&\leq \sum_k \max_i \left( \max_{\mu_i \in \Pi_i^{\text{det}}} \mathbb{E}_{\mu_{-i} \sim \pi^k_{-i}} \left[ \overline{V}_i^{k,\mu_i \times \mu_{-i}} \right] - V_i^{\pi^k} \right) \\
&= \sum_k \max_i \left( \mathbb{E}_{\mu \sim \pi^k} \left[ \overline{V}_i^{k,\mu} \right] - \mathbb{E}_{\mu \sim \pi^k} \left[ V_i^{\mu} \right] \right),
\end{aligned}
$$
(4)

where the final equality uses the fact that $\pi^k$ is a Nash equilibrium of the normal-form game defined by $(\overline{V}_1^k, \ldots, \overline{V}_n^k)$ as described in Subroutine 1. By Jensen's inequality and Azuma-Hoeffding inequality,

$$\sum_k \max_i \left( \mathbb{E}_{\mu \sim \pi^k} \left[ \overline{V}_i^{k,\mu} \right] - \mathbb{E}_{\mu \sim \pi^k} \left[ V_i^\mu \right] \right)$$

$$\leq \sum_k \mathbb{E}_{\mu \sim \pi^k} \left[ \max_i \left( \overline{V}_i^{k,\mu} - V_i^\mu \right) \right]$$

$$\leq \sum_k \max_i \left( \overline{V}_i^{k,\mu^k} - V_i^{\mu^k} \right) + \tilde{\mathcal{O}}(H\sqrt{K})$$

$$= \sum_k \max_i \left( \max_{\hat{\theta} \in \mathcal{B}^k} V_i^{\mu^k}(\hat{\theta}) - V_i^{\mu^k} \right) + \tilde{\mathcal{O}}(H\sqrt{K})$$

$$\leq H \sum_k \max_{\hat{\theta} \in \mathcal{B}^k} \left\| \mathbb{P}_{\hat{\theta}}^{\mu^k} - \mathbb{P}_{\theta^\star}^{\mu^k} \right\|_1 + \tilde{\mathcal{O}}(H\sqrt{K}),$$

where the last equality uses the definition of $\overline{V}^k$ and the last inequality uses the fact that the reward is an $H$-bounded function of the trajectory. Finally, we complete the proof by using the second relation in Theorem 19, which upper bounds $\sum_k \max_{\hat{\theta} \in \mathcal{B}^k} \left\| \mathbb{P}_{\hat{\theta}}^{\mu^k} - \mathbb{P}_{\theta^\star}^{\mu^k} \right\|_1$ by $\tilde{\mathcal{O}}\left( \frac{S^2 AO}{\alpha^2} \sqrt{K(S^2A + SO)} \times \text{poly}(H) \right)$.

**Coarse correlated equilibrium**   When we choose EQUILIBRIUM in Subroutine 1 to be CCE, the proof is exactly the same as for Nash equilibrium, except that the last equality in Equation (4) becomes "no larger than" by the definition of CCE.

**Correlated equilibrium**   When we choose EQUILIBRIUM in Subroutine 1 to be CE, by the definition of CE-regret,

$$\begin{aligned}
\text{Regret}_{\text{CE}}(K) &= \sum_k \max_i \left( \max_{\phi_i} V_i^{(\phi_i \diamond \pi_i^k) \odot \pi_{-i}^k} - V_i^{\pi^k} \right) \\
&= \sum_k \max_i \left( \max_{\phi_i} \mathbb{E}_{\mu \sim \pi^k} \left[ V_i^{(\phi_i \diamond \mu_i) \times \mu_{-i}} \right] - V_i^{\pi^k} \right) \\
&\leq \sum_k \max_i \left( \max_{\phi_i} \mathbb{E}_{\mu \sim \pi^k} \left[ \overline{V}_i^{k,(\phi_i \diamond \mu_i) \times \mu_{-i}} \right] - V_i^{\pi^k} \right) \\
&= \sum_k \max_i \left( \mathbb{E}_{\mu \sim \pi^k} \left[ \overline{V}_i^{k,\mu} \right] - \mathbb{E}_{\mu \sim \pi^k} \left[ V_i^\mu \right] \right),
\end{aligned} \tag{5}$$

where the second equality uses the definition of strategy modification, and the final equality uses the fact that $\pi^k$ is a CE of the normal-form game defined by $(\overline{V}_1^k, \ldots, \overline{V}_n^k)$ as described in Subroutine 1. The remaining steps are the same as of the proof for Nash-regret.

## C.2   Proof of Theorem 9

In this section, we prove Theorem 9 with a specific polynomial dependency as stated in the following theorem.

**Theorem 20.** (Total suboptimality of multi-step OMLE-Equilibrium) *Under Assumption 2, there exists an absolute constant $c$ such that for any $\delta \in (0, 1]$ and $K \in \mathbb{N}$, Algorithm 2 with*

$$\beta = c \left( H(S^2A + SO)\log(SAOHK) + \log(K/\delta) \right)$$

*and* EQUILIBRIUM *being one of {Nash, CCE, CE} satisfies (respectively) that with probability at least $1 - \delta$,*

$$\text{Regret}_{\{\text{Nash,CCE,CE}\}}(k) \leq \tilde{\mathcal{O}}\left( \frac{S^2 A^{3m-2}}{\alpha^2} \sqrt{k(S^2A + SO)} \times \text{poly}(H) \right) \qquad \text{for all } k \in [K],$$

*where the regret is computed for policy $\pi^1, \ldots, \pi^k$.*

---

**Algorithm 5** multi-step OMLE-Equilibrium

---

1: **Initialize:** $\mathcal{B}^1 = \{\hat{\theta} \in \Theta : \min_h \sigma_S(\hat{\mathbb{M}}_h) \geq \alpha\}$, $\mathcal{D} = \{\}$
2: **for** $k = 1, \ldots, K$ **do**
3:    compute $\pi^k =$ Optimistic_Equilibrium($\mathcal{B}^k$) and sample $\mu^k$ from $\pi^k$
4:    **for** $h = 0, \ldots, H - m$ **do**
5:        execute policy $\mu_{1:h}^k \circ \text{uniform}(\mathcal{A})$ to collect a trajectory $\tau^{k,h}$
          then add $(\mu_{1:h}^k \circ \text{uniform}(\mathcal{A}), \tau^{k,h})$ into $\mathcal{D}$
6:    update

$$\mathcal{B}^{k+1} = \left\{\hat{\theta} \in \Theta : \sum_{(\pi,\tau)\in\mathcal{D}} \log \mathbb{P}_{\hat{\theta}}^{\pi}(\tau) \geq \max_{\theta'\in\Theta} \sum_{(\pi,\tau)\in\mathcal{D}} \log \mathbb{P}_{\theta'}^{\pi}(\tau) - \beta \right\} \bigcap \mathcal{B}^1$$

---

The proof of Theorem 20 follows basically the same arguments as in the undercomplete setting, except that we replace Algorithm 4 with Algorithm 5 [6] and Theorem 19 with Theorem 21 in the first two steps. And the third step is exactly the same. To avoid noninformative repetitive arguments, here we only state Algorithm 5 and Theorem 21, while one can directly verify all the proofs in Section C.1 still hold after we make the aforementioned replacements.

**Theorem 21.** ([26]) *Under Assumption 2 and the same choice of $\beta$ as in Theorem 9, with probability at least $1 - \delta$, Algorithm 5 satisfies that for **all** $k \in [K]$ and **all** $\theta^1 \in \mathcal{B}^1, \ldots, \theta^K \in \mathcal{B}^K$*

- $\theta^\star \in \mathcal{B}^k$ ,

- $\sum_{t=1}^{k} \|\mathbb{P}_{\theta^t}^{\mu^t} - \mathbb{P}_{\theta^\star}^{\mu^t}\|_1 \leq \tilde{\mathcal{O}}\left(\frac{S^2 A^{3m-2}}{\alpha^2} \sqrt{k(S^2 A + SO)} \times \text{poly}(H)\right)$.

We remark that Theorem 21 follows directly from instantiating the analysis of multi-step optimistic MLE (Appendix F in [26]) on Algorithm 5.

## D   Proofs for Playing against Adversarial Opponents

### D.1   Proof of Theorem 12

In this section, we prove Theorem 12 with a specific polynomial dependency as stated in the following theorem.

**Theorem 22.** (Regret of OMLE-Adversary) *Under Assumption 1, there exists an absolute constant $c$ such that for any $\delta \in (0, 1]$ and $K \in \mathbb{N}$, Algorithm 3 with $\beta = c\left(H(S^2 A + SO)\log(SAOHK) + \log(K/\delta)\right)$ satisfies that with probability at least $1 - \delta$,*

$$\sum_{t=1}^{k}\left(\max_{\tilde{\pi}_1} \min_{\tilde{\pi}_2} V_1^{\tilde{\pi}_1 \times \tilde{\pi}_2} - V_1^{\pi^t}\right) \leq \tilde{\mathcal{O}}\left(\frac{S^2 AO}{\alpha^2}\sqrt{k(S^2 A + SO)} \times \text{poly}(H)\right) \quad \textit{for all } k \in [K].$$

To begin with, for the same reasons as explained in Section C.1.2, we can directly instantiate the guarantees for optimistic MLE (Appendix E in [26]) on Algorithm 3 and obtain:

**Theorem 23.** ([26]) *Under Assumption 1 and the same choice of $\beta$ as in Theorem 12, with probability at least $1 - \delta$, Algorithm 3 satisfies that for **all** $k \in [K]$ and **all** $\theta^1 \in \mathcal{B}^1, \ldots, \theta^K \in \mathcal{B}^K$*

- $\theta^\star \in \mathcal{B}^k$ ,

- $\sum_{t=1}^{k} \|\mathbb{P}_{\theta^t}^{\pi^t} - \mathbb{P}_{\theta^\star}^{\pi^t}\|_1 \leq \tilde{\mathcal{O}}\left(\frac{S^2 AO}{\alpha^2}\sqrt{k(S^2 A + SO)} \times \text{poly}(H)\right)$.

---

[6]We remark that in each episode $k$ of Algorithm 2, we sample the random seed $\omega$ used in $\pi^k$ *only once* and then combine $\pi^k(\omega, \cdot)$ with random actions starting from different in-episode steps to collect multiple trajectories (Line 4-5 in Algorithm 2). Therefore, Algorithm 5 and Algorithm 2 are *equivalent* for the same reasons as explained in Section C.1.1.

Now conditioning on the two relations in Theorem 23 being true, we have

$$\sum_k \left( \max_{\hat\pi_1} \min_{\hat\pi_{-1}} V_1^{\hat\pi_1 \times \hat\pi_{-1}} - V_1^{\pi_1^k \times \pi_{-1}^k} \right)$$

$$= \sum_k \left( \max_{\hat\pi_1} \min_{\hat\pi_{-1}} V_1^{\hat\pi_1 \times \hat\pi_{-1}}(\theta^\star) - \max_{\hat\theta \in \mathcal{B}^k} \max_{\hat\pi_1} \min_{\hat\pi_{-1}} V_1^{\hat\pi_1 \times \hat\pi_{-1}}(\hat\theta) \right)$$

$$+ \sum_k \left( \max_{\hat\theta \in \mathcal{B}^k} \max_{\hat\pi_1} \min_{\hat\pi_{-1}} V_1^{\hat\pi_1 \times \hat\pi_{-1}}(\hat\theta) - V_1^{\pi_1^k \times \pi_{-1}^k}(\theta^\star) \right)$$

$$\theta^\star \in \mathcal{B}^k \quad \le \sum_k \left( \max_{\hat\theta \in \mathcal{B}^k} \max_{\hat\pi_1} \min_{\hat\pi_{-1}} V_1^{\hat\pi_1 \times \hat\pi_{-1}}(\hat\theta) - V_1^{\pi_1^k \times \pi_{-1}^k}(\theta^\star) \right)$$

$$\text{by the definition of } \pi_1^k \quad = \sum_k \left( \max_{\hat\theta \in \mathcal{B}^k} \min_{\hat\pi_{-1}} V_1^{\pi_1^k \times \hat\pi_{-1}}(\hat\theta) - V_1^{\pi_1^k \times \pi_{-1}^k}(\theta^\star) \right)$$

$$\le \sum_k \left( \max_{\hat\theta \in \mathcal{B}^k} V_1^{\pi_1^k \times \pi_{-1}^k}(\hat\theta) - V_1^{\pi_1^k \times \pi_{-1}^k}(\theta^\star) \right)$$

$$\text{reward per episode} \in [0, H] \quad \le H \sum_k \max_{\hat\theta \in \mathcal{B}^k} \left\| \mathbb{P}_{\hat\theta}^{\pi^k} - \mathbb{P}_{\theta^\star}^{\pi^k} \right\|_1$$

$$\text{Theorem 23} \quad \le \tilde{\mathcal{O}} \left( \frac{S^2 AO}{\alpha^2} \sqrt{k(S^2 A + SO)} \times \text{poly}(H) \right).$$

## D.2 Proof of Theorem 11

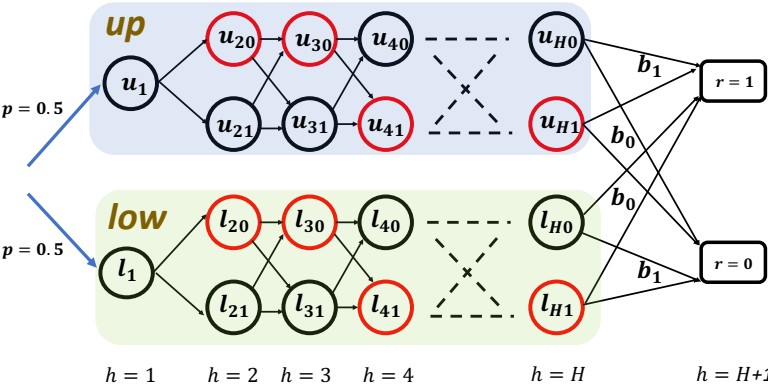

Figure 1: hard instance for Theorem 11.

The hard instance is best illustrated by Figure 1 where we sketch the transition dynamics of the POMG. Below we elaborate the construction based on Figure 1.

- **States and actions.** Each circle and rectangle in Figure 1 represents a state. Each player has two actions denoted by $\mathcal{A} = \{a_0, a_1\}$ and $\mathcal{B} = \{b_0, b_1\}$ respectively.

- **Observations.** The max-player can always directly observe the current latent state. The min-player observes the same dummy observation $o_{\text{null}}$ in any black circle while directly observes the current state in any red circle and any rectangle. It is easy to verify by definition $\min_h \sigma_{\min}(\mathbb{O}_h) = 1$ because the emission structure is a bijection between the joint observation space and the latent state space.

- **Reward.** Only the upper rectangle emits an observation containing reward 1 for the max-player (thus reward $-1$ for the min-player). All other states emit observations with zero reward.
- **Transitions.** At the beginning of each episode, the environment starts from $u_1$ or $l_1$ uniformly at random. The transition dynamics from step 1 to step $H$ only depend on the actions of the max-player while in the final step ($H \to H+1$) the transitions are determined by the min-player. Formally,
    - When the environment is in the **upper half** of the POMG: for each step $h \in [H-1]$, the environment will transition to $u_{h+1,0}$ if the max-player takes action $a_0$ and transition to $u_{h+1,1}$ if the max-player takes action $a_1$. At step $H$, the agent will transition to the upper rectangle if the min-player takes action $b_1$ and transition to the lower one if the min-player picks $b_0$.
    - When the environment is in the **lower half** of the POMG: for each step $h \in [H-1]$, the environment will transition to $l_{h+1,0}$ if the max-player takes action $a_0$ and transition to $l_{h+1,1}$ if the max-player takes action $a_1$. At step $H$, the agent will transition to the upper rectangle if the min-player takes action $b_0$ and transition to the lower one if the min-player picks $b_1$.

**Min-player's optimal strategy.** It is direct to see the min-player's optimal strategy is to take action $b_0$ in the upper half of the POMG and action $b_1$ in the lower half, at step $H$. This stratety will lead to zero-reward for the max-player. However, implementing this strategy requires the min-player to infer which half the environment is in from her observations, which is possible only when the environment has visited some red circles in the first $H$ steps. This is because the min-player directly observes the current state in red circles while observes the same observation $o_{\text{null}}$ in all black circles.

**Max-player's optimal strategy.** To prevent the min-player from discovering which half the environment currently lies in, the max-player's optimal strategy is to avoid visiting any red circles.

**Hardness.** However, hardness happens if (a) the max-player cannot access the observations of the min-player and (b) for each $h \in \{2, \ldots, H\}$, we uniformly at random pick one of $\{u_{h0}, u_{h1}\}$ and one of $\{l_{h0}, l_{h1}\}$ to be red circles, and set the remaining ones to be black. From the perspective of the max-player, she cannot directly tell which state is red or black because (a) the difference between black circles and red circles only appear in the min-player's observations, and (b) the max-player cannot see what the min-player observes. As a result, the only useful information for the max-player to figure out which circles are red is the action picked by the min-player in the final step.

Now suppose the min-player will play the optimal strategy when she knows which half the environment is in, and pick action $b_0$ when she does not. In this case, for the max-player, identifying all the red circles is as hard as learning a bandit with $\Omega(2^H)$ arms where only one arm has reward $1/2$ and all other arms has reward 0. Therefore, by using standard lower bound arguments for bandits, we can show the max-player's cumulative rewards in the first $K = \Theta(2^H)$ episodes is 0 with constant probability. In comparison, the optimal strategy, which avoids visiting all red circles, can collect $K/3$ rewards with high probability. As a result, we obtain the desired $\Omega(\min\{2^H, K\})$ regret lower bound for competing against the Nash value.

### D.3 Playing against adversary in multi-step weakly-revealing POMGs is hard

In this section, we prove that competing with the max-min value is statistically hard even if (i) the POMG is two-player zero-sum and satisfies Assumption 2 with $m = 2$ and $\alpha = 1$, (ii) the opponent keeps playing a fixed action, and (iii) the player can directly observe the opponents' actions and observations.

**Theorem 24.** *Assume the player can directly observe the opponents' actions and observations. For any $L, k \in \mathbb{N}^+$, there exist (i) a two-player zero-sum POMG of size $S, A, O, H = \mathcal{O}(L)$ and satisfying Assumption 2 with $m = 2$ and $\alpha = 1$, and (ii) an opponent who keeps playing a fixed action $\hat{a}_2$, so that with probability at least $1/2$*

$$\sum_{t=1}^{k} \left( \max_{\tilde{\pi}_1} \min_{\tilde{\pi}_2} V_1^{\tilde{\pi}_1 \times \tilde{\pi}_2} - V_1^{\pi_1^t \times \hat{a}_2} \right) \geq \Omega\left(\min\{2^L, k\}\right),$$

*where $\pi_1^t$ is the policy played by the learner in the $t^{\text{th}}$ episode.*

*Proof.* The hard instance is constructed as following:

- **States and actions**: There are four states: $p_0, p_1$ and $q_0, q_1$. Each player has two actions, denoted by $\{a_0, a_1\}$ and $\{b_0, b_1\}$ respectively.

- **Emission and reward**: There are three different observations: $o_{\mathrm{dummy}}$, $o_1$ and $o_0$. At step $h \in [H-1]$, $p_1$ and $p_0$ emit the same observation $o_{\mathrm{dummy}}$. At step $H$, $p_1$ emits $o_1$ while $p_0$ emits $o_0$. Regardless of $h$, $q_0$ always emits $o_0$ and $q_1$ always emits $o_1$. Importantly, all players share the same observation. The reward function is defined so that $r(o_{\mathrm{dummy}}) = r(o_0) = 0$ and $r(o_1) = 1$ for the max-player. Since the game is zero-sum, the reward function for the min-player is simply $-r(\cdot)$.

- **Transition**: Let $x_1, \ldots, x_h$ be a binary sequence sampled independently and uniformly at random from standard Bernoulli distribution. At step $h = 1$, the POMG always starts from state $p_1$. For each step $h \in [H-1]$:
  - If the current state is $p_i$, then the environment will transition to $p_1$ if and only if $i = 1$, the max-player plays action $a_{x_h}$, and the min-player plays $b_0$. Otherwise, if the min-player plays $b_1$, then the environment will transition to $q_i$. Otherwise, the environment will transition to $p_0$.
  - If the current state is $q_i$, the next state will be $q_1$ regardless of players' actions.

We have the following observations:

- If the min-player keeps playing $b_0$, then from the perspective of the max-player the POMG essentially reduces to a multi-arm bandit problem with $2^{H-1}$ arms because in this case the only useful feedback for the max-player is the reward observed at step $H$.

- The max-min (Nash) value is equal to 1, which is attained when the max-player picks $a_{x_h}$ at step $h$ with probability 1.

- The 2-step emission-action matrix at each step $h \in [H-1]$ is rank 4 and has minimum singular value no smaller than 1, because we can always exactly identify the current state (for step $h \in [H-1]$) by the current-step observation and the next-step observation if the min-player picks action $b_1$ in the current step.

Based on the first two observations above, we immediately obtain a $\Theta(\min\{2^H, k\})$ lower bound for competing with the max-min (Nash) value. Using the third observation, we know the POMG is 2-step weakly revealing with $\alpha = 1$, which completes the proof. $\qquad\square$