# OpenReview forum: "Sample-Efficient Reinforcement Learning of Partially Observable Markov Games"
_NeurIPS.cc/2022/Conference — NeurIPS 2022 Accept_

### Official Review · Reviewer_7FnW · 2022-07-09

**Rating:** 6
**Confidence:** 3
**Soundness:** 4 excellent
**Presentation:** 3 good
**Contribution:** 4 excellent

**Summary:**

This paper is the first attempt to theoretically analyze the general-sum Markov games under partial observability. The contributions of this paper are as follows:
- It identify a rich subclass of POMGs, namely, the weakly revealing POMGs, which is statistically tackable (this is necessary as the POMGs are intackable in general as shown by the hardness results);
- It proposes an algorithm that combines the optimism-in-face-of-uncertainty principle and MLE;
- The proposed algorithm is shown to be efficient in the self-play setting, meaning that we can control all the agents' policies;
- The results are extened to the adversarial setting, which is novel and also interesting.

I believe that the topic of this paper is very interesting and is very relevant to the ML and RL community.

**Questions:**

See weakness part.

**Limitations:**

yes

**Strengths And Weaknesses:**

Strengths:
Identifying a rich sub-class of tackable POMGs:
RL problems under partial observability are known to be statistically intractable in general, even in the single-agent case, mainly due to the non-Markovian nature and resulting curse of history. However, the hardness is in a worst-case fashion. Then, it is natural to identify certain natural (well-motivated) structural assumption that empowers efficient learning.

Instead of assuming the perfect-recall structure of Imperfect Information Extensive-Form Games (IIEFG), this paper considers the weakly revealing POMGs, extended from the single-agent scenario. To me, the two assumptions are natural. Intuitively, the assumption ensures that a short window of recent observations contains certain amount of information of the latent states, whose idea is natural and common under the partial observability [1-3]. Interestingly, in terms of the challenges from multi-agent scenario, the paper shows that this condition only requires to be true for the joint observations, which is new to my knowledge. I believe that this paper provides enough insights and is a good starting point for the study of POMGs.

Algorithmic novelty:
I find that the proposed algorithms are quite clean, following the standard optimism-in-face-of-uncertainty principle. Although the algorithm is not computationally efficeint, but it seems this is a general problem on the literature, so I think this is acceptable.

The key design of the confidence set is extended from the single-agent counterpart [4] that contains the models whose likelihood is roughly consistent on the historical data (in the sense that itis close to MLE), which is also natural. The use of MLE seems to be extended from the recent result in single-agent setting [4]. But the game nature requires extra efforts to deal with, which accounts for the use of subroutines in different settings. This again is new and is complementary to the single-agent counterpart.

Weakness:
While I am satisfactory with the paper, I still have some minor comments and questions.

1 I think some examples or hardness results are helpful to motivate the assumptions used in this paper, for a clearer presentation.

2 The adversarial setting is interesting but it seems that the paper only considers the Assumption 1, which is true only for undercomplete case (O>=S). I am wondering whether efficient learning is possible for the overcomplete case?

Overall, I enjoyed reading this paper and I think it contains enough insight, and interesting and non-trivial results, as compared to the single-agent works.

[1] Du, S., Krishnamurthy, A., Jiang, N., Agarwal, A., Dudik, M., & Langford, J. (2019, May). Provably efficient RL with rich observations via latent state decoding. In International Conference on Machine Learning (pp. 1665-1674). PMLR.
[2] Efroni, Yonathan, Chi Jin, Akshay Krishnamurthy, and Sobhan Miryoosefi. "Provable reinforcement learning with a short-term memory." arXiv preprint arXiv:2202.03983 (2022).
[3] Golowich, Noah, Ankur Moitra, and Dhruv Rohatgi. "Learning in Observable POMDPs, without Computationally Intractable Oracles." arXiv preprint arXiv:2206.03446 (2022).
[4] Qinghua Liu, Alan Chung, Csaba Szepesvari, and Chi Jin. When is partially observable reinforcement
learning not scary? arXiv preprint, 2022a.

---

> ### Author Response · Authors · 2022-08-02
> **Response to Reviewer 7FnW**
>
> Thank you for your positive and detailed reviews. Please see our responses below.
>
> ---
>
> **Q**. I think some examples or hardness results are helpful to motivate the assumptions used in this paper, for a clearer presentation.
>
> **A**. Thank you for the suggestion. For POMDPs (i.e., POMGs with a single player),  [KAL16, JKKL20] proved that without making any assumption, even finding a 1/2-optimal policy requires exponential samples in general. And the hard instances they constructed are those POMDPs where there exist two different latent state mixtures with the same observation distributions, i.e., $\exists \mu_1, \mu_2 \in \Delta_S$ s.t. $\mu_1\neq\mu_2$ but $\mathbb{O}\mu_1=\mathbb{O}\mu_2$. One can verify that such problems appear if and only if the rank of the emission matrix is smaller than the number of states. Therefore, it is natural to rule out such pathological examples, where sample-efficient learning is impossible, by assuming the rank of the emission matrix is equal to the number of states. And the $\alpha$-weakly revealing condition is simply a robust version of this full-rank condition.
>
> Extending the weakly revealing conditions from POMDPs to POMGs leads to two natural candidates: either (a) joint observations or (b) individual observations are required to weakly reveal the state information. This paper shows the former (the weaker assumption) suffices to guarantee tractability.
>
> ---
>
> **Q**. The adversarial setting is interesting but it seems that the paper only considers the Assumption 1, which is true only for undercomplete case ($O\ge S$). I am wondering whether efficient learning is possible for the overcomplete case?
>
> **A**. The multi-step weakly revealing condition is in general not compatible with the adversarial setting because it requires all players to jointly take certain explorative sequences in order to acquire information about the latent dynamics, which is impossible in the adversarial setting where the learner only controls a single player. As a result, if the opponents never take the explorative action sequence, then learning could be extremely hard for the learner in the worst case. Following this intuition, we can prove that there exists an exponential regret lower bound for playing against adversarial opponents in multi-step weakly revealing POMGs even if (i) the POMG is two-player zero-sum and satisfies Assumption 2 with $m=2$ and $\alpha=1$, (ii) the opponent keeps playing a fixed action, and (iii) the player can directly observe the opponents’ actions and observations. We have added this hardness result in the revised version (Appendix E.4).
>
> ---
>
>  **Reference**
>
>   [KAL16] PAC reinforcement learning with rich observations. A. Krishnamurthy, A. Agarwal, and J. Langford.
>
>  [JKKL20] Sample-efficient reinforcement learning of undercomplete POMDPs. C. Jin, S. M. Kakade, A. Krishnamurthy, and Q. Liu.

---

### Official Review · Reviewer_HtwJ · 2022-07-09

**Rating:** 6
**Confidence:** 2
**Soundness:** 3 good
**Presentation:** 3 good
**Contribution:** 3 good

**Summary:**

The paper identifies a rich subclass of  Partially Observable Markov Games (POMGs), weakly revealing POMGs, and proposes algorithms for finding approximate Nash equilibria for them. The algorithm is proven to be sample efficient in this kind of game.

**Questions:**

In Algorithm 1, Why is the output strategy uniformly sampled from past strategies?

**Limitations:**

None.

**Strengths And Weaknesses:**

Strengths:
1. The newly identified weakly revealing POMGs is interesting and practical.
2. The proposed algorithms are intuitive and interesting. The theoretical results seem sound.
3. The paper is well written, and the notation and definitions are clear.

Weaknesses:
1. Corollary 8 and Corollary 10, "with probability at least 1/2", I do not think a probability of at least 1/2 is enough.
2. As stated in the paper, "Algorithm 1 is computationally inefficient in general due to the steps of optimistic value estimation and equilibrium computation". By the way, this does not harm the significance of the paper, and thanks to the authors for declaring the weakness.
3. Lack of empirical results. Some experiments may help the readers to understand the algorithms better.
4. line 90, $\mathbb{T}$ is not defined, while $\mathbb{P}$ is not introduced.
5. line 113, the definition of $\pi_i$ is not a bit informal.

---

> ### Author Response · Authors · 2022-08-02
> **Response to Reviewer HtwJ**
>
>
> Thank you for your positive reviews. Please see our responses below.
>
> ---
> **Q.** Corollary 8 and Corollary 10, "with probability at least 1/2", I do not think a probability of at least 1/2 is enough.
>
> **A.**  There is a standard argument to boost the success probability from $1/2$ to $1-\delta$. Specifically, we can achieve so by the following three steps: (i) first sample $L:=\mathcal{O}(\log(1/\delta))$ policies, denoted as $\\{\hat{\pi}^{l}\\}\_{l\in[L]}$, with replacement from $\\{\pi^k\\}\_{k\in[K]}$, (ii) then test the suboptimality of each $\hat{\pi}^l$, and (iii) pick the one with the lowest suboptimality. To explain how/why this scheme works, below let us take Corollary 8 and Nash equilibrium as an example, and the other cases follow almost the same:
> 1.  By Bernstein inequality, we have with probability at least $1-\delta/2$, there exists at least one $\hat{\pi}^l\in\\{\hat{\pi}^{l}\\}_{l\in[L]}$ which is $\epsilon$-optimal.
>   2.  By definition 1, estimating the suboptimality of $\hat{\pi}^l$ is equivalent to estimating $V\_{i}^{\dagger,\hat{\pi}^l\_{-i}}-V^{\hat{\pi}^l}\_i$ for each player $i\in[n]$. Now let us consider a fixed $(l,i)\in[L]\times[n]$. First note that $V^{\hat{\pi}^l}\_i$ can be easily estimated to high accuracy by executing policy $\hat{\pi}^l$ for $\tilde{\mathcal{O}}(H^2/\epsilon^2)$ episodes. To estimate $V\_{i}^{\dagger,\hat{\pi}^l\_{-i}}$, we can rerun Algorithm 1 by only modifying Line 3 in the following way: all players but player $i$ follow $\hat{\pi}^l$ (i.e., $\pi^k\_{-i} = \hat{\pi}^l\_{-i}$) and player $i$ follows the most optimistic policy against $\hat{\pi}^l\_{-i}$ (i.e., $\pi^k\_{i} = \arg\max_{\pi\_i\in\Pi\_i^{\rm det}} \max\_{\hat{\theta}\in\mathcal{B}^k}V_{i}^{\pi_i\times\hat{\pi}^l_{-i}}(\hat{\theta})$). By doing so with the same choice of $K$ as Corollary 8, we can show $(\sum_{k=1}^K V_i^{\pi^k})/K$ (which is easy to estimate from the collected trajectories) is an $\epsilon$-accurate estimate of  $V_{i}^{\dagger,\hat{\pi}^l_{-i}}$ with high probability. By repeating this process for all $(l,i)\in[L]\times[n]$, we can estimate the suboptimality of $\hat{\pi}^l$ to precision $\epsilon$ for all $l\in[L]$ with probability at least $1-\delta/2$.
> 3. Putting all pieces together, we can find an $\mathcal{O}(\epsilon)$-optimal policy with probability at least $1-\delta$.
>
> Note that the above process will only increase the overall sample complexity by a multiplicative factor of $\mathcal{O}(N\mathrm{polylog}(N/\delta))$. Therefore, our sample complexity guarantee remains polynomial in all parameters.
>
> ---
>
> **Q.** As stated in the paper, "Algorithm 1 is computationally inefficient in general due to the steps of optimistic value estimation and equilibrium computation". By the way, this does not harm the significance of the paper, and thanks to the authors for declaring the weakness. Lack of empirical results. Some experiments may help the readers to understand the algorithms better.
>
> **A.** Thank you for the suggestion. Even for POMDPs (i.e., POMGs with a single player) with known model parameters, planning (i.e., computing a near-optimal policy) is computationally hard [PT87, VLB12]. Therefore, it remains highly unclear under what conditions we can design and how to design computationally efficient (implementable) algorithms for POMDPs and POMGs. We believe this is an important open problem worth further investigation.
>
> ---
>
> **Q.**  Line 90, $\mathbb{T}$ is not defined, while $\mathbb{P}$ is not introduced. line 113, the definition of $\pi_i$ is a bit informal.
>
> **A.**  Thank you for spotting typos.   $\mathbb{P}$ should be $\mathbb{T}$ in Line 93, 94 and 109.
> Throughout this paper, we always use $\mathbb{T}$ to denote transitions and $\mathbb{P}$ to denote probability.
> We have fixed the typos in the revision. To our best knowledge, our definition of $\pi_i$ is rigorous, which uses the standard terminology in probability theory.
>
> ---
>
> **Q.**  In Algorithm 1, Why is the output strategy uniformly sampled from past strategies?
>
> **A.**  This is the consequence of the online-to-batch conversion: because Theorem 7 only guarantees the average suboptimality of $\\{\pi^k\\}\_{k\in[K]}$ is low, which in general cannot imply the last policy $\pi^K$ is near-optimal. Moreover, we believe there exist POMG instances where the last policy $\pi^K$ in Algorithm 1 can be highly suboptimal. Therefore, Algorithm 1 does not output the last policy but instead samples $\pi^{\rm out}$ uniformly at random from $\\{\pi^k\\}\_{k\in[K]}$ (by Markov inequality, $\pi^{\rm out}$ will be near-optimal with constant probability).
>
> ---
>
> **Reference**
>
>   [PT87]  The complexity of Markov decision processes. C. H. Papadimitriou and J. N. Tsitsiklis.
>
>  [VLB12]  On the computational complexity of stochastic  controller optimization in POMDPs. N. Vlassis, M. L. Littman, and D. Barber.

---

> > ### Comment · Reviewer_HtwJ · 2022-08-08
> > **Thanks for the response.**
> >
> > I am satisfied with the response to my questions, and I will not change the rating.

---

### Official Review · Reviewer_9qur · 2022-07-11

**Rating:** 5
**Confidence:** 3
**Soundness:** 4 excellent
**Presentation:** 2 fair
**Contribution:** 3 good

**Summary:**

This paper considers a Multi-Agent Reinforcement Learning problem where participants have private observations that are only partially available to others. In a setting where a central learner can control all players' actions and aims to achieve an equilibrium, the authors develop algorithms to approximate equilibria with a sample-efficiency bound. Moreover, in a setting where a single player aims to optimize their reward against adversarial opponents, they also create an algorithm to achieve a sublinear upper regret bound under a weak definition of the regret.

**Questions:**

1. Can I have more interpretation of why the regret in Section 5 is used. Besides, are there any practical applications of this new version of regret?

2. There might be some typos and unclear parts. For example, in the definition of the POMG tuple, does $\mathbb{T}$ means $\mathbb{P}$? Moreover, in Definition 6, I wonder what the feasible set of strategy modifications is. Does it contain all maps from the deterministic action set to itself?

**Limitations:**

I think there is no potential negative social impact of the work.

**Strengths And Weaknesses:**

Strengths:

1. This is the first work that develops a sample efficiency algorithm to learn a large class of Partially Observable Markov Games. Moreover, the polynomial sample complexity is strong.

2. The motivation of their algorithms is clear.

Weakness:

1. For the setting with adversarial opponents, it is better to give more intuition of the reason to use their new regret.

2. Their algorithms seems to be time-consuming.

3. There might be some typos and unclear parts that may affect understanding. Please see bullet 2 in the question section for more details.

---

> ### Author Response · Authors · 2022-08-02
> **Response to Reviewer 9qur**
>
> Thank you for your positive reviews. Please see our responses below.
>
> ---
>
> **Q.** Can I have more interpretation of why the regret in Section 5 is used. Besides, are there any practical applications of this new version of regret?
>
> **A.** The main reason that we did not use the standard version of regret  ($\max\_{\tilde{\pi}\_i}\sum_{k=1}^K(V\_i^{\tilde{\pi}\_i\times\pi^k\_{-i}}-V\_i^{\pi^k})$) is that there exists exponential regret lower bound for competing with the best response in hindsight even in fully observable two-player zero-sum Markov games, which are special cases of POMGs satisfying Assumption 1 with $\alpha=1$. This means that no algorithm can achieve sublinear regret in terms of the standard notion of regret in POMGs without introducing further assumptions. We briefly explained this point in Line 322-326 on Page 9 and more details about the hardness results can be found in [LWJ22] (e.g., their Theorem 2 and 4).
>
> The regret used in this paper is weaker than the standard one and corresponds to competing with the max-min value of the game, which is the highest value player $i$ can achieve if all other players jointly play against her. For example, in two-player zero-sum POMGs where the max-min value is equal to the value of Nash equilibria by the minimax theorem, Theorem 12 guarantees that the learner can collect cumulative rewards almost as high as always playing the Nash strategy.  A more concrete toy example is rock-scissor-paper, where our regret guarantee implies that the learner will only lose the game for a number of times sublinear in the total number of gameplay. Similar criterion was also used in practical AI system design [e.g, BS18]  where the goal of the algorithm is to collect cumulative rewards as high as the Nash value (i.e., the regret considered in this paper) instead of competing with the best-response in hindsight (i.e., the standard regret notion).
> Finally, we point out that our regret notion has also been commonly used even in the literature of MGs [e.g., BT02, TWYS20].
>
> ---
>
> **Q.**  Their algorithms seem to be time-consuming.
>
> **A.**  Even for POMDPs (i.e., POMGs with a single player) with known model parameters, planning (i.e., computing a near-optimal policy) is computationally hard [PT87, VLB12]. Therefore, it remains highly unclear under what conditions we can design and how to design computationally efficient (implementable) algorithms for POMDPs and POMGs. We believe this is an important open problem worth further investigation.
>
> ---
>
> **Q.** There might be some typos and unclear parts. For example, in the definition of the POMG tuple, does $\mathbb{T}$  mean $\mathbb{P}$? Moreover, in Definition 6, I wonder what the feasible set of strategy modifications is. Does it contain all maps from the deterministic action set to itself?
>
> **A.**  Thank you for spotting typos.   $\mathbb{P}$ should be $\mathbb{T}$ in Line 93, 94 and 109.
> Throughout this paper, we always use $\mathbb{T}$ to denote transitions and $\mathbb{P}$ to denote probability.
> We have fixed the typos in the revision. In Definition 6, the feasible set of strategy modifications include all the strategy modifications defined in Line 138-145 on Page 4. We would like to emphasize that the set of strategy modifications considered in this paper include all maps from the set of deterministic policies to itself, which is strictly larger than the set of maps defined on action sets.
>
> ---
>  **Reference**
>
>   [PT87]  The complexity of Markov decision processes. C. H. Papadimitriou and J. N. Tsitsiklis.
>
>   [BT02] R-MAX - A General Polynomial Time Algorithm for Near-Optimal Reinforcement Learning. R. I. Brafman, M. Tennenholtz.
>
>
>  [VLB12]  On the computational complexity of stochastic  controller optimization in POMDPs. N. Vlassis, M. L. Littman, and D. Barber.
>
>  [BS18] Superhuman AI for heads-up no-limit poker: Libratus beats top professionals. N. Brown and T. Sandholm.
>
>  [TWYS20] Online Learning in Unknown Markov Games. Y. Tian, Y. Wang, T. Yu, S. Sra.
>
>
>  [LWJ22] Learning markov games with adversarial opponents: Efficient  algorithms and fundamental limits. Q. Liu, Y. Wang, and C. Jin.

---

> > ### Comment · Reviewer_9qur · 2022-08-07
> > **Rebuttal Acknowledgement**
> >
> > Thank you very much for the reply! My questions are addressed, and I will keep my original score.

---

### Official Review · Reviewer_QFwE · 2022-07-18

**Rating:** 4
**Confidence:** 5
**Soundness:** 3 good
**Presentation:** 3 good
**Contribution:** 2 fair

**Summary:**

This paper considers the Partially Observed (general-sum) Markov Game (POMG) model, and proposes a sub-linear regret online learning algorithm that achieves a game theoretic equilibria. The algorithm is based on optimism ideas, and an assumption of weak revealing POMGs is made (which was first made in [27,37], and is quite strong actually - it implies full observability for point state distributions. The proposed algorithm achieves regret that grows as \sqrt{K}, K=number of episodes.

**Questions:**

Q.1 Is there a way to design decentralized learning algorithms?

**Limitations:**

No limitations mentioned.

**Strengths And Weaknesses:**

Strengths: The paper is nicely written, easy to understand and addresses a difficult problem.

Weaknesses:
1. I think this paper is just a bit beyond incremental, builds as it does on [27] and [37].
2. There is a more fundamental issue whether these results are interesting at all: in a game setting, there are two types of results that are interesting: (i) offline (centralized or decentralized) algorithms that find the equilibrium, and (ii) online decentralized algorithms that minimize the regret w.r.t. the equilibrium. This paper proposes online  centralized algorithms for POMGs, which means in a game setting individual players can't really learn in a decentralized manner, i.e., learning of the equilibria is really happening via  centralized algorithm, which is hardly interesting. Actually, once you assume that an equilibria of the game can be computed by some algorithm, the results in this paper are hardly surprising building as they do on results for POMDPs in [27] and [37]. Thus, I would say the problem formulation in this paper is not interesting enough.
3. Any way to do an empirical evaluation of your proposed algorithms.

---

> ### Author Response · Authors · 2022-08-02
> **Response to Reviewer QFwE (1/2)**
>
> Thank you for your review. Please see our responses below.
>
> ---
>
> **Q.**  The algorithm is based on optimism ideas, and an assumption of weak revealing POMGs is made (which was first made in [27,37], and is quite strong actually - it implies full observability for point state distributions.
>
> **A.**  We would like to clarify that the weakly revealing condition does NOT imply full observability for point state distributions. Specifically, even if the weakly revealing condition holds and the model parameters are known, it is still impossible to identify the latent states (that is, point state distributions) from observations and actions. The weakly revealing condition only requires that two different (mixtures of) states would emit two different distributions of joint observations, which is very natural in partially observable systems.
>
> We also want to emphasize that even in POMDPs, the weakly revealing condition is arguably one of the weakest conditions which can guarantee sample-efficient learning in the exploration setting. Before the weakly revealing condition was introduced, most previous works for learning POMDPs either make *additional* strong reachability assumptions [e.g., ALA16, GDB16] or only consider very special emission structures like block MDPs [e.g., KAL16, JKALS17] (special cases of weakly revealing POMDPs with $\alpha\ge 1/\sqrt{O}$).
>
> Finally, to demonstrate the generality of the weakly revealing conditions, we show that imperfect-information extensive-form games (IIEFGs) can be easily represented as 1-weakly revealing POMGs with $\prod\_i X\_i$ ($X\_i$ denotes the number of player $i$’s information sets) states, the same action space and the same observation space. As a result, all the algorithms and theoretical guarantees developed in this paper immediately apply to learning IIEFGs with polynomial sample-efficiency guarantees. We have added the detailed reduction for IIEFGs in the revised version (Appendix B).
>
> ---
>
> **Q.**  I think this paper is just a bit beyond incremental, builds as it does on [27] and [37].
>
> **A.**   We would like to re-emphasize that POMGs is a natural, very general, and important class of problems in reinforcement learning, and this paper provides the first line of sample-efficient results for POMGs which we view as significant. Prior works [27, 37] only address POMDPs, and do not address at all the game-theoretical parts of problems such as collaboration v.s. competition, or compete against adversarial opponents (as in our Section 5). The multiagent perspective of POMGs requires many new treatments for the solution concepts comparing to [27, 37] ranging from adjusting the basic definition of "weakly revealing" to incorporate local observations of each player to designing new planning algorithms to find various equilibria. We would also like to point the reviewer to Section 1.1 "technical novelty" and the bullet points within for more details.
>
>
> ---
>
>
> **Q.** Is there a way to design decentralized learning algorithms?
>
> **A**.  The standard approach to designing  decentralized learning algorithms for normal-form games or IIEFGs is by using no-regret learning algorithms. However, Markov Game is fundamentally more challenging, as it has been proved that any algorithm against adversarial opponents incur exponential regret in the worst case (see Theorem 2 and 4 in [LWJ22]), thus no-regret learning with polynomial dependency on all problem-dependent parameters is not possible.
>
> To our best knowledge, the only class of online decentralized algorithms for MGs is V-learning [JLWY21], which in its current form only applies to tabular and fully observable problems.
> However, it remains highly unclear how to extend their results to either the function approximation setting or the partially observable setting. We believe this is an important open problem worth further investigation.

---

> > ### Author Response · Authors · 2022-08-02
> > **Response to Reviewer QFwE (2/2)**
> >
> >
> > **Q.**  There is a more fundamental issue whether these results are interesting at all: in a game setting, there are two types of results that are interesting: (i) offline (centralized or decentralized) algorithms that find the equilibrium, and (ii) online decentralized algorithms that minimize the regret w.r.t. the equilibrium. This paper proposes online centralized algorithms for POMGs, which means in a game setting individual players can't really learn in a decentralized manner, i.e., learning of the equilibria is really happening via centralized algorithm, which is hardly interesting.
> >
> > **A.** To clarify your terminology, do you mean “the model is known” by offline and “the model is unknown and interaction is needed” by online? If so, our response is as below:
> > 1.  Our paper studies the online centralized setting which is strictly more general than the offline centralized setting  because sampling is straightforward when the model is known. Since the reviewer suggested the offline centralized setting is interesting, then we believe the more general online centralized setting should also be interesting.
> >    2. Online centralized algorithms are very practical in the context of e.g., training competitive game-playing AI. Online centralized algorithms have achieved huge success in modern reinforcement learning for finding equilibria strategy in highly complex game environments. For example, the training process of AlphaStar [VBC+19] (an AI agent designed for playing StarCraft) and OpenAI Five [OpenAI18] (an AI agent designed for playing Dota) are both online and centralized, which features centralized coordination of all players and millions of online game-play between them.
> >   3. Online decentralized algorithms are fundamentally challenging. See the answer to the previous question.
> >
> > ---
> >
> > **Q.** Actually, once you assume that an equilibrium of the game can be computed by some algorithm, the results in this paper are hardly surprising building as they do on results for POMDPs in [27] and [37].
> >
> > **A.**  We would like to clarify that we did NOT assume an equilibrium of the game can be computed by some algorithm. Instead we carefully designed an optimistic planning algorithm (Subroutine 1) that effectively addresses both the game-theoretical aspects of the problem and the exploration challenge under partial observability. This subroutine is even distinct from the standard techniques for learning MGs, which we believe is of sufficient technical novelty. Moreover, we want to point out that our paper also provides a comprehensive study of the setting of playing against adversarial opponents. These results are completely new, and unique to the multiagent setup.
> >
> > ---
> >
> > **Q.** Any way to do an empirical evaluation of your proposed algorithms.
> >
> > **A.**   Even for POMDPs (i.e., POMGs with a single player) with known model parameters, planning (i.e., computing a near-optimal policy) is computationally hard [PT87, VLB12]. Therefore, it remains highly unclear under what conditions we can design and how to design computationally efficient (implementable) algorithms for POMDPs and POMGs. We believe this is an important open problem worth further investigation.
> >
> > ---
> >
> > **Reference**
> >
> >   [PT87]  The complexity of Markov decision processes. C. H. Papadimitriou and J. N. Tsitsiklis.
> >
> >  [VLB12]  On the computational complexity of stochastic  controller optimization in POMDPs. N. Vlassis, M. L. Littman, and D. Barber.
> >
> > [ALA16] Reinforcement learning of POMDPs using spectral methods. K.  Azizzadenesheli, A. Lazaric, and A. Anandkumar.
> >
> > [GDB16] A PAC RL algorithm for episodic
> > POMDPs. Z. Guo, S. Doroudi, and E. Brunskill.
> >
> >   [KAL16] PAC reinforcement learning with rich observations. A. Krishnamurthy, A. Agarwal, and J. Langford.
> >
> >    [JKALS17] Contextual decision processes with low Bellman rank are PAC-learnable. N. Jiang, A. Krishnamurthy, A. Agarwal, J. Langford, and R. E. Schapire.
> >
> > [VBC+19] Grandmaster level in StarCraft II using multi-agent reinforcement learning. O. Vinyals, I. Babuschkin, W. M.  Czarnecki,...,D. Silver.
> >
> > [OpenAI18] OpenAI Five. OpenAI.
> >
> > [JLWY21] V-Learning -- a simple, efficient, decentralized algorithm for multiagent RL. C. Jin, Q. Liu, Y. Wang, and T. Yu.
> >
> >  [LWJ22] Learning markov games with adversarial opponents: Efficient algorithms and fundamental limits. Q. Liu, Y. Wang, and C. Jin.

---

> > ### Comment · Reviewer_QFwE · 2022-08-08
> > **Still not convinced**
> >
> > 1. Ok, fair point about weakly revealing condition being more general than fully observable.
> >
> > 2.  I issue is not that POMGs are not important, but rather that once an MDP (or POMDP) problem is solved, solving zero-sum games just requires replacing the max operator with a minmax. And almost all the results go through without  too much difficulty.
> >
> > 3. The difficulty in Markov games (from MDPs) and hence POMGs (from POMDPs) arises when we try to design online decentralized learning algorithms (not the same as centralized learning of decentralized policies).
> >
> > Overall, I think the results don't go far enough.

---

> > > ### Author Response · Authors · 2022-08-09
> > > **Disagree on the comments for technical contribution**
> > >
> > > **Q**. Once an MDP (or POMDP) problem is solved, solving zero-sum games just requires replacing the max operator with a minmax. And almost all the results go through without too much difficulty.
> > >
> > > **A**. We respectfully but strongly disagree with this comment:
> > > 1. This paper considers multiplayer general-sum setting, which is neither two-player nor zero-sum. We also consider the solution concepts of CE/CCE, which is beyond minmax.
> > > 2. This paper also considers the setting of playing against **adversarial opponents**, which does not have a single-agent counterpart. The resulting algorithm is online, but not centralized. This comment (as well as the discussion of online centralized vs decentralized) completely ignores these interesting results.
> > > 3. Even within the two-player zero-sum POMGs, simply replacing max operator with minmax would NOT work. In fact, if we directly replace the max in optimistic planning of OMLE algorithm in [1] with minimax, it does not even result in a valid algorithm, as OMLE is maximizing over models (transition + emission), which does not decouple between two players (thus can't be minmax). Maybe the reviewer was considering the *fully observable* setting, where one can replace max with minmax for the Bellman equations in MGs. However, in the *partially observable* setting, we can't directly use Bellman equations (neither does our algorithm), since we can't easily estimate transition/emission which involve unobserved latent states. Not to mention that the Bellman equation involves value functions which have exponential complexity in POMGs (as value depends on entire history or continuous belief states), and during all these we still need to address exploration. **The techniques we use to extend from POMDPs to POMGs are fundamentally different from the standard techniques of extending from MDPs to MGs.** It's critical to design our optimistic planning algorithm (Subroutine 1) that effectively addresses *both* the game-theoretical aspects of the problem and the exploration challenge under partial observability. Our planning algorithm is *distinct* from the standard techniques for learning MGs.
> > >
> > > For more details about our technical contribution, please refer to Section 1.1 "technical novelty" and also our previous response.
> > >
> > > [1] Q. Liu, A. Chung, C. Szepesvari, and C. Jin. When is partially observable reinforcement learning not scary? arXiv preprint, 2022.

---

### Meta-Review · Area_Chair_Hiaf · 2022-08-26

**Recommendation:** Accept
**Confidence:** Certain

**Metareview:**

While most reviewers were positive about the paper, one reviewer expressed concerns on the assumption and the technical novelty of the paper. After reading the discussion, the AC believes that the main assumption of weakly revealing is standard. This is commonly used in control and learning in HMMs/PSRs (we just cannot learn efficiently without these type of assumptions). The AC is also satisfied by the authors response to the technical novelty. The results presented in this paper is certainly more than just replacing max by min-max in the Bellman equation.

In summary, the AC would recommend for acceptance in this case.

**Award:**

No

---

### Decision · Program_Chairs · 2022-09-14

Accept